# Molecular tuning of farnesoid X receptor partial agonism

Daniel Merk[1,6], Sridhar Sreeramulu[2,6], Denis Kudlinzki[2,3,4], Krishna Saxena[2,3,4], Verena Linhard[2], Santosh L. Gande[2,3,4], Fabian Hiller[2], Christina Lamers [1], Ewa Nilsson[5], Anna Aagaard [5], Lisa Wissler [5], Niek Dekker[5], Krister Bamberg [5], Manfred Schubert-Zsilavecz[1] & Harald Schwalbe[2,3,4]

The bile acid-sensing transcription factor farnesoid X receptor (FXR) regulates multiple metabolic processes. Modulation of FXR is desired to overcome several metabolic pathologies but pharmacological administration of full FXR agonists has been plagued by mechanism-based side effects. We have developed a modulator that partially activates FXR in vitro and in mice. Here we report the elucidation of the molecular mechanism that drives partial FXR activation by crystallography- and NMR-based structural biology. Natural and synthetic FXR agonists stabilize formation of an extended helix α11 and the α11-α12 loop upon binding. This strengthens a network of hydrogen bonds, repositions helix α12 and enables co-activator recruitment. Partial agonism in contrast is conferred by a kink in helix α11 that destabilizes the α11-α12 loop, a critical determinant for helix α12 orientation. Thereby, the synthetic partial agonist induces conformational states, capable of recruiting both co-repressors and co-activators leading to an equilibrium of co-activator and co-repressor binding.

[1] Institute of Pharmaceutical Chemistry, Goethe University, Frankfurt 60348, Germany. [2] Center for Biomolecular Magnetic Resonance (BMRZ), Institute for Organic Chemistry and Chemical Biology, Goethe University, Frankfurt 60438, Germany. [3] German Cancer Consortium (DKTK), Heidelberg 69120, Germany. [4] German Cancer Research Center (DKFZ), Heidelberg 69120, Germany. [5] Innovative Medicines and Early Development Biotech Unit, AstraZeneca, Gothenburg 43183, Sweden. [6] These authors contributed equally: Daniel Merk, Sridhar Sreeramulu. Correspondence and requests for materials should be addressed to D.M. (email: merk@pharmchem.uni-frankfurt.de) or to H.S. (email: schwalbe@em.uni-frankfurt.de)

The ligand-activated transcription factor farnesoid X receptor (FXR, NR1H4)[1,2] acting as cellular sensor for bile acids is a crucial metabolic regulator and liver protector[3]. As a member of the nuclear receptor (NR) superfamily it translates ligand signals into changes in gene expression and affects a variety of genes involved in bile acid, glucose and lipid metabolism. For this central biological role, FXR has gained considerable attention as drug target for severe liver disorders[4] and metabolic diseases[5]. The first in class FXR agonist obeticholic acid (OCA)[6] has revealed very promising effects of pharmacological FXR activation in non-alcoholic steatohepatitis (NASH), in primary biliary cholangitis (PBC) and in diabetes. OCA has recently been approved by the FDA for PBC treatment and is currently investigated in late stage clinical development for NASH[7,8] despite its adverse effects that appear partly arising from FXR over-activation. Elevated cholesterol levels have been reported from clinical trials[7,9] with OCA that can be correlated with the finding that complete FXR activation blocks bile acid biosynthesis and hinders metabolic cholesterol degradation as consequence. Importantly, conversion to bile acids accounts for approximately one third of cholesterol elimination. Partial FXR activation, therefore, appears as a valuable strategy to avoid mechanism-based side effects of FXR targeting and recently, promising pre-clinical and phase 1 clinical data of a partial FXR agonist have been reported[10]. Partial agonists induce activation of the NR with reduced activation efficacy compared to agonists which translates to modest modulation of FXR-regulated gene expression[11]. However, the molecular mode of action by which partial agonism is established and its structural determinants in terms of ligand binding and conformational changes in the FXR ligand binding domain (LBD) have remained elusive.

NRs share a common architecture of an N-terminal ligand-independent activation domain followed by a DNA-binding domain (DBD), a flexible hinge region and a ligand-binding domain composed of twelve α-helices (α1-α12). Especially the C-terminal α12, the so-called activation function 2 (AF-2) of the LBD, has been identified as crucial for NR activation since stabilisation of α12 and concomitant binding to the core of the LBD provides a surface for co-activator binding[12].

Previous studies have already evaluated general molecular mechanisms of NR activation. Observations in early co-crystal structures of NR-LBDs revealing differences in the position of α12 for liganded and unliganded (apo) states, have established the common model of a mouse-trap mechanism for NR activation where α12 was thought to be positioned away from the core LBD in apo-state. Agonist binding would then alter its position to be bound to the core LBD[13]. Later studies (reviewed in[14]) have used various techniques to revise this model and have suggested that the terminal LBD region is dynamic in apo-state and that α12 is only formed upon agonist binding. This contradicted the model of two (active and inactive) positions of a permanently stable α12 and already indicates that more than a single active conformation of NR-LBDs may exist.

According to current understanding, agonistic NR ligands promote stabilisation of α12 and binding to the core LBD to induce co-activator recruitment while antagonists stabilise an inactive state with unordered AF-2 of the transcription factor that binds to co-repressors. However, this commonly accepted mechanism of α12 stabilisation and subsequent co-activator recruitment upon agonist binding explains FXR agonism and antagonism fairly well while it does not offer a sound concept concerning partial FXR agonism, so far.

Differential binding modes within the FXR ligand binding site of FXR agonists OCA and MFA-1[15], and differential effects of FXR agonists GW4064, CDCA and fexaramine on FXR regulated gene expression[16] already indicated several years ago, that FXR activation is more complex than just switched on or off. Later studies revealed that FXR ligands can differ in their recruitment profiles of various co-activators[17,18], a characteristic that has also been observed for other NRs such as peroxisome proliferator-activated receptors[19,20]. Moreover, ivermectin[18] and structural analogues[21] were identified as specific FXR modulators that despite recruiting the NR co-repressor NCoR-2 to the FXR-LBD induced FXR-mediated reporter activity and gene expression, and recent results also point to an allosteric mechanism[22] of FXR activation by small molecules. However, the molecular basis of the differential modes of FXR modulation has not been deciphered.

In order to gain a better understanding of the molecular basis of partial FXR activation, we utilised the potent partial FXR agonist 1 (Fig. 1) for structural investigations (X-ray and NMR) on the FXR-LBD that deciphered the molecular mechanism of FXR activation. In particular, we established an NMR-based technique to observe FXR's activation state in solution allowing a correlation of equilibria between conformational populations with concomitant biological effects. Our studies draw a comprehensive picture of FXR modulation by ligands that promote our understanding of this crucial physiological and pathophysiological transcriptional regulator.

## Results

**Development and characterisation of partial FXR agonist 1.** Previously, we have conducted structure-activity-relationship[23–25] studies on FXR ligands yielding potent partial agonists. Compound 1 (DM175, Fig. 1a) shares the anthranilamide scaffold but differs in the geometry of the benzoic acid moiety from previously reported analogues. In a full-length transactivation assay based on a FXR response element from the promoter region of bile salt export protein (BSEP), 1 activated FXR with a transactivation efficacy of $21.1 \pm 0.5\%$ compared to FXR agonist GW4064[26] at $3\,\mu M$ concentration and with an $EC_{50}$ value of $0.35 \pm 0.06\,\mu M$ (Fig. 1b). In line with its partial agonistic activity, 1 also revealed partial FXR antagonistic potency and repressed GW4064-induced FXR activity to $26 \pm 2\%$ activation ($IC_{50} = 10.9 \pm 0.2\,\mu M$). In vitro characterisation of the previously reported FXR modulator ivermectin revealed a partial agonistic/antagonistic profile, as well. Evaluation of 1 in a hybrid Gal4-FXR reporter gene assay confirmed its partial FXR agonism/antagonism (Fig. 1c). Profiling of the fatty acid mimetic[27] 1 on related lipid binding NRs as well as the membrane bile acid receptor TGR5 (Fig. 1d) revealed selectivity except weak off-target activities on PPARγ and RARα. Moreover, 1 was non-toxic (Fig. 1e) and metabolically stable (Fig. 1f), and possessed preferable aqueous solubility (133 ng/mL) and lipophilicity (logP 2.0) in light of its fatty acid mimetic[27] structure. In mice (male C57BL6/J), a single oral dose of 1 (10 mg/kg) resulted in a favourable pharmacokinetic profile with high bioavailability and a half-life of 2.1 h confirming suitability of 1 for in vivo studies (Fig. 1g).

When FXR-expressing liver (HepG2) or intestinal (HT-29) cells were incubated with 1, partial induction (40%) of FXR regulated genes relative to the endogenous FXR-agonist CDCA (2) was observed (Fig. 1h). Accordingly, the indirect FXR target gene CYP7A1 was repressed by about 40% compared to CDCA confirming the partial agonistic activity of 1 in a more physiological setting than the reporter gene assay. Application of 1 to mice confirmed that 1 behaved as partial agonist in vivo as well since a partial induction of the FXR target gene SHP and a partial repression of CYP7A1 compared to mice receiving CDCA were observed (Fig. 1i). Together, this promising in vitro and in vivo profile proved 1 as suitable for investigations on the molecular mechanism of partial FXR activation.

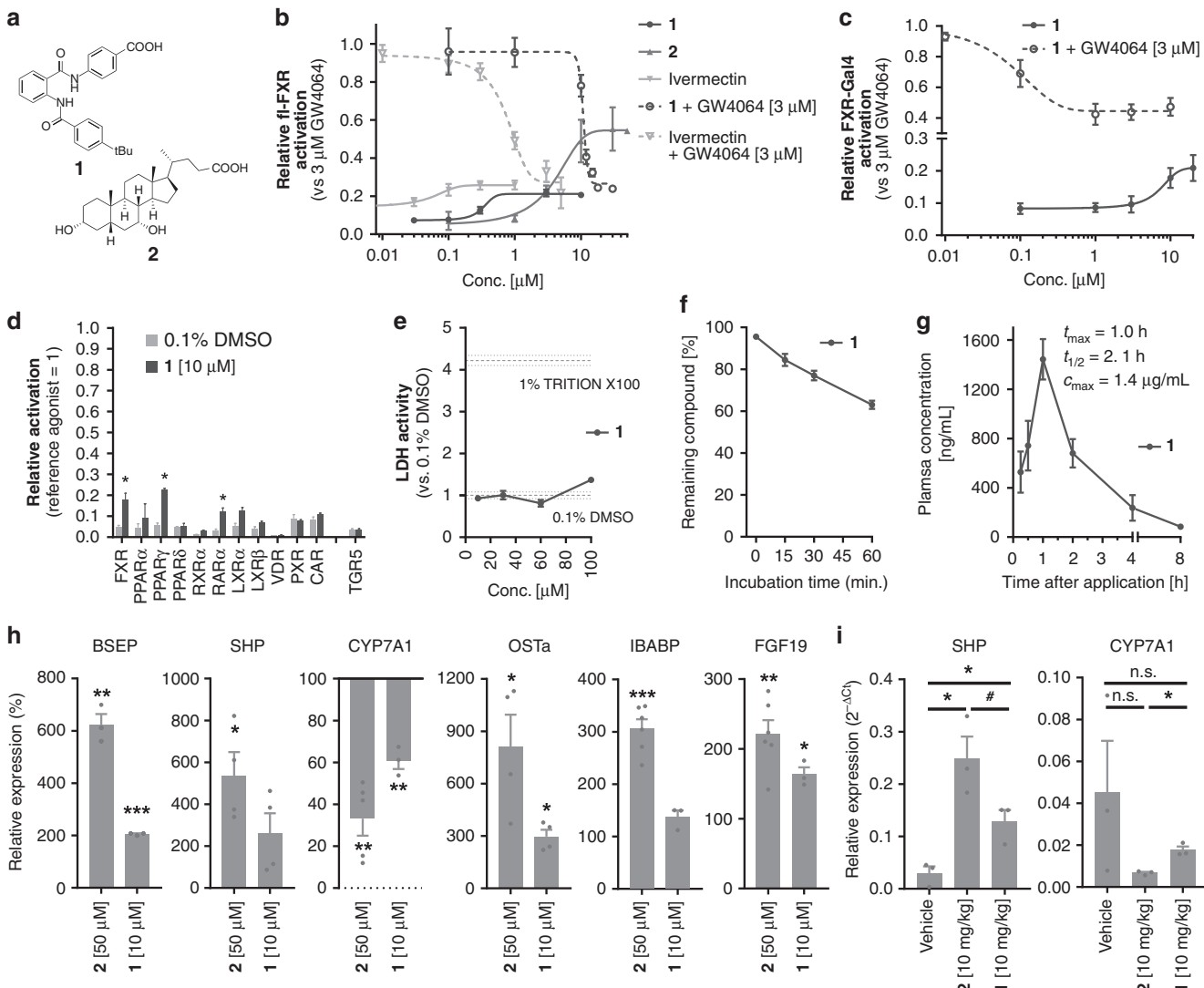

**Fig. 1** Structure and pharmacological profile of partial FXR agonist **1**. **a** Constitution of **1** and endogenous agonist CDCA (**2**). **b** Dose-response curves of partial agonist **1**, endogenous agonist **2** and ivermectin in a BSEP response element driven full-length FXR reporter gene assay. **1** partially activates FXR with an $EC_{50}$ value of 0.35 ± 0.06 μM and 21.1 ± 0.5% relative efficacy compared to the synthetic agonist GW4064 (3 μM). **1** also partially represses GW4064-induced FXR activation with an $IC_{50}$ value of 10.9 ± 0.2 μM to 26 ± 2% relative activation. CDCA (**2**) behaves as weakly potent ($EC_{50}$ = 4.0 ± 0.2 μM) agonist. Ivermectin has partial FXR agonistic activity ($EC_{50}$ = 0.060 ± 0.007 μM, 26 ± 1% rel. activation) and suppresses GW4064-induced FXR activation ($IC_{50}$ = 0.80 ± 0.16 μM, 22 ± 6% rel. act.). Results are mean ± SD, n = 3. **c** Partial agonistic activity of **1** was also observed in a hybrid Gal4-FXR reporter gene assay. **d** Profiling of **1** on NRs in hybrid reporter gene assays. Results are mean ± SEM, n = 3. **e** In vitro toxicity of **1** in HepG2 cells: **1** exhibits no acute toxicity in vitro up to 100 μM concentration. Results are mean ± SEM, n = 4. **f** In vitro metabolism analysis: **1** comprises good stability versus microsomal degradation with >60% of the parent compound remaining after 60 min. incubation. Results are mean ± SEM, n = 4. **g** Pharmacokinetic profile of **1** in C57BL/6j mice: With high oral bioavailability and a half-life of more than 2 h, **1** is suitable for in vivo studies. Results are mean ± SEM, n = 3. **h** Profiling of the effects of **1** on FXR regulated gene expression: Compared to endogenous FXR agonist CDCA, **1** partially induced FXR regulated genes in HepG2 cells (BSEP, SHP, CYP7A1, OSTα) and in HT-29 cells (IBABP, FGF19). Results are mean ± SEM, n = 3. **i** Hepatic mRNA levels of FXR-regulated genes upon treatment with **1** in mice: In mouse livers, **1** caused partial induction of SHP and partial repression of CYP7A1 compared to CDCA confirming its partial agonistic properties in vivo. Results are mean ± SEM, n = 3. Statistical significance was analysed by two-sided student's t-test. #$p < 0.1$, *$p < 0.05$, **$p < 0.01$, ***$p < 0.001$ vs. 0.1% DMSO or as indicated. Source data are provided as a Source Data file

**X-ray structures of FXR-CDCA and FXR-partial agonist**. As first step towards understanding the molecular determinants of FXR partial agonism, we solved the co-crystal structures (Fig. 2, Supplementary Table 1) of the FXR-LBD (residues 244–472) in complex with physiological agonist **2** (6HL1) and with partial agonist **1** (4QE8). The FXR-LBD complex containing **1** crystallised as a dimer. Both ligand-bound FXR-LBDs are composed of 12 α-helices with bound α-helical NCoA-2740-2752 peptide and adopt the typical three-layer α-helical sandwich arrangement that resembles most NR structures[28]. The 1.65 Å high-resolution FXR/

CDCA (**2**) structure comprises residues E244-Q472. Notably, the loop region linking α4/5 and α6 (K339-P341) is invisible suggesting flexibility and dynamics of this loop induced by ligand binding. The 2.6 Å FXR/**1** structure comprises the identical range as FXR/CDCA and also lacks the loop region between α4/5 and α6 (P341-S342). Additionally, the electron density of the loop region connecting α11 and α12 (V456-H459, AF-2 loop) is ambiguous or invisible, due to induced flexibility by binding of **1**. The temperature factors (Supplementary Fig. 1) reveal differences in flexibility within helices α2 and α6 for both structures. This

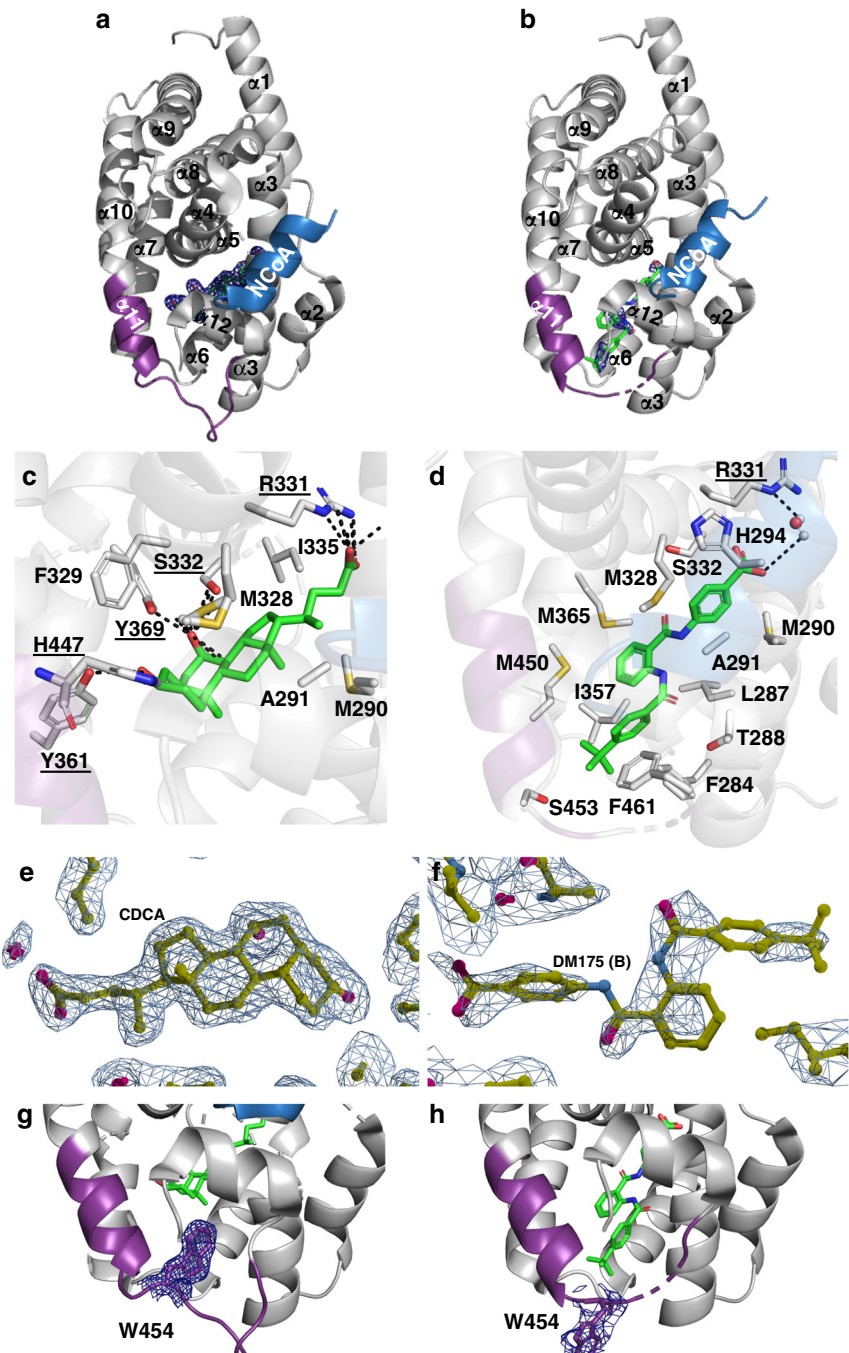

**Fig. 2** Binding modes of physiological FXR agonist CDCA and partial agonist **1** to the FXR-LBD. **a** CDCA bound to FXR. The co-activator (NCoA) is highlighted in blue. **b** Compound **1** bound to FXR. **c, d** Key residues involved in the interaction with CDCA and compound **1**, respectively. **e, f** 2Fc-Fo omit maps (contour level 1.0 σ) for CDCA and compound **1** (chain B), respectively bound to the FXR-LBD. Chain B was considered for all interpretation due to preferable electron density for the ligand. Ligand density for chain A is poor due to twinning of the crystal. Additionally, the electron density of the loop region connecting α11 and α12 (V456-H459, AF-2 loop) in chain B is ambiguous or invisible, due to induced flexibility by binding of **1**. **g, h** Compared to CDCA-bound FXR, binding of the partial agonist **1** causes an outward movement of W454 by 12 Å (2Fc-Fo omit maps, contour level 1.0 σ)

flexibility is slightly more pronounced in the complex of **2**. The Ω-loop connecting α1 and α3 is considerably destabilised by binding of **1** compared to CDCA (**2**).

The structures solved here containing **2** (6HL1, Fig. 2a) and **1** (4QE8, Fig. 2b) aligned well with very low average r.m.s.d. with structures containing obeticholic acid (OCA, PDB-ID: 1OSV[29]) or GW4064 (PDB-ID: 3DCT[30]). Positions and orientations of all α-helices are very well conserved independently of the bound ligand and expectedly, all ligands occupy a similar binding site.

Minor shifts of the N-terminal part of α7 and the C-terminal part of α11 result from different ligand sizes and binding modes. Additionally, there are apparent differences in loop (L) regions connecting α1-α2 (L:α1-α2), α5-α6 (L:α5-α6), α6-α7 (L:α6-α7) and α11-α12 (L:α11-α12) which are differently orientated indicating that residues of these regions are involved in or affected by compound binding. Binding of CDCA (**2**) disturbs L: α5-α6 folding and binding of **1** destabilises L:α5-α6 as well as L: α11-α12.

**Table 1 Activities of FXR agonists and partial agonists on wild-type FXR and mutants W454Y and W454A**

|  | wild-type EC$_{50}$ (max. rel. activation) | W454Y EC$_{50}$ (max. rel. activation) | W454A EC$_{50}$ (max. rel. activation) |
| --- | --- | --- | --- |
| GW4064 | 0.07 ± 0.03 µM (108 ± 8%) | 0.020 ± 0.003 µM (100 ± 2%) | 0.010 ± 0.003 µM (102 ± 4%) |
| CDCA | 4.0 ± 0.2 µM (55 ± 1%) | inactive at 50 µM | inactive at 50 µM |
| Ivermectin | 0.060 ± 0.007 µM (26 ± 1%) | 0.03 ± 0.01 µM (34 ± 1%) | inactive at 1 µM |
| partial agonist **1** | 0.35 ± 0.05 µM (21.1 ± 0.5%) | 0.005 ± 0.003 µM (31 ± 2%) | 0.003 ± 0.001 µM (25.0 ± 0.4%) |

Wild-type FXR and mutants W454Y and W454A were studied in reporter gene assays in HeLa cells using the FXR response element from the promoter region of bile salt export protein (BSEP) to govern reporter expression. CDCA (**2**) and ivermectin were tested up to the highest non-toxic concentration as indicated. GW4064 (3 µM) was used as reference and defined as 100% activation. Results are the mean ± SEM, $n = 3$. Source data are provided as a Source Data file.

The ligand binding site in FXR is defined by α-helices 2, 3, 5, 6, 7, 11 and 12. The agonistic ligands CDCA (**2**), OCA and GW4064 occupy the same binding pocket but show differences in their residue specific interaction modes. CDCA (**2**) is bound via conserved sidechain interactions involving R331, S332, Y361, Y369, H447 and non-bonded contacts (Fig. 2c). The steroidal FXR agonists OCA (1OSV) and 3-deoxy-CDCA (1OT7[29], with exception of O3 interactions) are similarly stabilised, while synthetic ligands such as GW4064 and **1** are differently bound. GW4064 binding is mainly mediated by 15 non-bonded interactions, two polar contacts of the carboxyl group with R331 and one with the backbone of M265. Partial agonist **1** is exclusively bound by 15 direct non-bonded interactions and one water mediated H-bond to R331 (Fig. 2d). W454 located on L:α11-α12 (AF-2 loop) is typically positioned inwards the hydrophobic binding pocket forming non-bonded contacts to CDCA or other agonists (Fig. 2e). Surprisingly, binding of **1** causes an outward movement of W454 by 12 Å driven through site occupation by the *tert*-butyl moiety designating the most striking difference in the structure of the complex with the partial agonist (Fig. 2g/h, Supplementary Fig. 2).

**Mutagenesis experiments of FXR-W454.** To probe the importance of residue W454 of FXR in mediating partial activation, we studied the activity of FXR agonists GW4064 and CDCA (**2**) as well as partial agonists **1** and ivermectin on FXR mutants W454Y and W454A. The $^1$H/$^{15}$N-HSQC spectra of the fully labeled W454Y and W454A mutant FXR-LBDs indicated that the protein fold is not affected by the mutations and that the mutations are structurally silent (Supplementary Fig. 3). In full-length FXR reporter gene assays with these mutants, residue 454, however, turned out to have major impact on FXR activation by ligands (Table 1, Supplementary Fig. 4). While activity of FXR agonist GW4064 was hardly affected by both mutations, endogenous FXR agonist CDCA (**2**) was inactive on both W454 mutants at 50 µM. This may result from a less hydrophobic environment when the highly lipophilic tryptophan residue is replaced by alanine or tyrosine. For partial agonists **1** and ivermectin, mutation of W454 to Y caused increased activation efficacy and for **1** also markedly enhanced potency. Mutation to A had a similar effect for partial agonist **1** but rendered ivermectin inactive. The lower EC$_{50}$ values of **1** on the mutants may result from reduced energetic penalty compared to the large movement of W454 in wild-type FXR and its exposure to the solvent.

**Analysis of FXR-LBD structures.** Because there is no major conformational change due to binding of different ligands, minor effects lead to differential FXR activation. By interaction of compounds with α3, α4/5, α10/11 or L:α11-α12 (the AF-2 loop), α12 is affected which in turn modifies the binding mode of the co-activator. In the agonistic conformation, the co-activator is bound by two conserved ionic interactions forming a charge clamp with K303 (α3) and E467 (α12) stabilising the dipolar NCoA-2 α-helix (N-terminus positively charged, C-terminus negatively charged). Hence, the surface for co-activator binding is crucially dependent on the position of α12. The agonistic structure of CDCA comprises the α12 helix and the entire loop connecting α11 and α12 (Fig. 3a) whereas in the partial agonistic structure, L:α11-α12 is invisible due to destabilisation and α12 occupies a slightly shifted position (Fig. 3c). In contrast, the antagonistic FXR structure (4WVD[18], co-crystallised with co-repressor instead of co-activator) containing ivermectin completely lacks α12 and L:α11-α12. Additionally, α11 is shortened and the structure comprises no co-activator but a co-repressor peptide (Fig. 3b).

Superposition of FXR-LBD structures (Fig. 3d) reveals that FXR ligands occupy different regions within the FXR ligand binding site. FXR agonists CDCA (**2**), GW4064 and fexaramine[15] bind close to helix 4 whereas the binding position of partial agonist **1** protrudes towards L:α11-α12 compared to agonistic ligands and ivermectin is shifted towards the Ω-loop causing partial destabilisation of the FXR-LBD thereby suggesting that it may adopt a "combined" binding mode similar to agonist and partial agonist. Of note, the FXR-LBD structure containing ivermectin is complexed with a fragment of nuclear receptor co-repressor 1 (NCoR-1) and, therefore, has to be considered as antagonistic conformation of the FXR-LBD although ivermectin biologically behaves as partial agonist on FXR. This observation, however, suggests that both the ligand and the co-regulator peptide affect FXR-LBD conformation.

In complex with agonists, FXR-LBD and co-activator typically interact via nine different H-bonds (involving K303, H313, E314, K321 and E467 of the FXR-LBD) and several non-bonded contacts. However, binding of **1** results in disruption of five of these H-bond interactions by shifting of α12 towards the co-activator peptide and thereby inducing an upward movement of NCoA-2 (Supplementary Fig. 5). These results suggest that partial FXR agonism is eventually explained by changes in the interaction between FXR-LBD and co-activator.

To gain more insights into the mode of interaction between LBD and co-activator, we investigated ~1300 NR-LBD structures deposited in the PDB of which ~550 contained a co-activator peptide. When we measured the distance between the C$_\alpha$ atoms of the two residues forming the charge clamp (Glu and Lys, Supplementary Fig. 6) we observed that this distance is highly flexible in absence of a co-activator peptide and ranges between 10 and 26 Å. However, in structures containing a co-activator peptide, the distance between the charge clamp residues is narrowed to a distance between 17 and 22 Å whatever kind of ligand (agonistic or partial agonistic) was bound to the LBD (Supplementary Fig. 6).

This analysis indicates that binding of the co-activator significantly contributes to stabilisation of the NR-LBD and thereby affects its conformation. Therefore, the frozen state of the co-crystal structure representing only one possible conformation

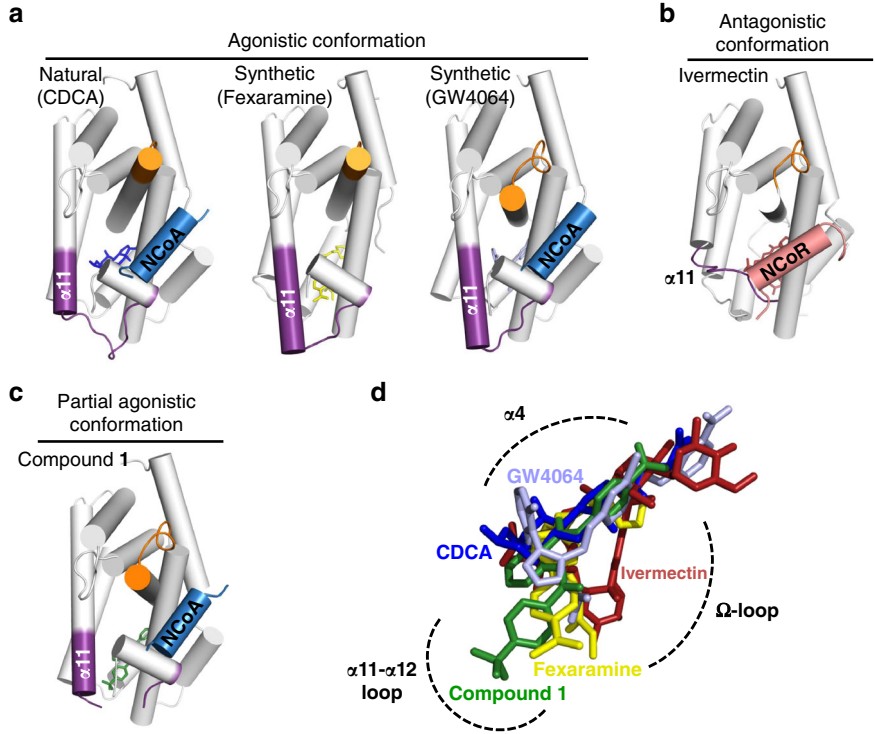

**Fig. 3** FXR-LBD – ligand/activator (blue)/repressor (light-red) interactions. **a** (left) The natural FXR agonist CDCA (blue) stabilises α11-α12 (AF2-loop) and α12 (AF2-helix) while the α11 (purple) is partially formed. **a** (middle, right) Binding of the synthetic agonists fexaramine (yellow) and GW4064 (light blue) has a profound effect and extends the length of the α11. **b** Binding of ivermectin, destabilises the AF2-loop and in particular the α11 and the Ω-loop. The ivermectin containing FXR-LBD structure is complexed with NCoR-1 and thus represents an antagonistic conformation not fully reflecting ivermectin's biological activity on FXR. **c** The partial agonist **1** by disturbing the AF2-loop influences the position of the AF2-helix and thereby affects position, orientation and binding of NCoA. In all structures **a–c** the α3-α4 loop is depicted in orange and the ligands are shown as sticks. **d** Overlay of the binding modes of different ligands. Key protein regions affected due to the ligand binding are highlighted with dashed-arcs

is not sufficient to evaluate the mechanism of partial activation. To overcome this limitation, we studied the interaction of the FXR-LBD with co-activator and co-repressor peptides in presence of ligands with different pharmacological activity in solution by NMR.

**NMR-based monitoring of FXR-ligand-co-regulator dynamics.** The $^1$H-$^{15}$N-HSQC of uniformly $^{15}$N-labeled FXR-LBD protein revealed only signals for about half the entire LBD indicating conformational mobility of the protein. Addition of FXR agonist CDCA lead to LBD stabilisation observable in shifting of signals. Further addition of co-activator peptide, however, was associated with significantly stronger stabilisation of the protein conformation indicated by appearance of almost all FXR-LBD signals and further shifts (Fig. 4). For the synthetic agonists GW4064 and fexaramine[16], equal effects were observed in the spectra. FXR antagonist guggulsterone induced no stabilisation of the FXR-LBD. In fact, only minor differences were observed in the spectra of FXR-LBD alone or in presence of guggulsterone. When we conducted the same experiment with the partial agonists **1** and ivermectin, stabilising effects were observed upon their addition to the FXR-LBD. Addition of the co-activator, however, induced no noticeable further stabilisation. Thus, the $^1$H-$^{15}$N-HSQC of the FXR-LBD confirmed strong effects of the co-activator peptide on the LBD conformation in presence of an agonist but indicated that interaction of FXR with partial agonists follows a different mechanism. To study this mode of partial FXR activation in solution and reveal differences between agonist and partial agonist, we used co-activator (NCoA, residues 740–752) and co-repressor (NCoR-1, residues 2256–2278) peptides each

containing one $^{13}$C/$^{15}$N-labeled leucine (Fig. 5a) together with unlabeled FXR-LBD protein for in-depth NMR studies.

To monitor recruitment and binding of the co-repressor, we first recorded the $^1$H/$^{13}$C-HSQC (Fig. 5a) of the labeled NCoR and NCoA peptides which each revealed two sharp signals for the Cβ of the labeled amino acid. Upon addition of the FXR-LBD to the labeled NCoR co-repressor peptide, the Cβ-signals disappeared due to severe line-broadening indicating that the peptide was bound to the FXR-LBD (Fig. 5b–d). When the physiological FXR agonist CDCA was subsequently added to this system (Fig. 5b, e), the NCoR Cβ signals partly reappeared suggesting that the co-repressor peptide was partly released due to agonist binding. Hence, an agonistic ligand induces an equilibrium of bound and free co-repressor. Finally, addition of the co-activator peptide fully restored the Cβ signals of the co-repressor indicating that binding of the co-activator to the FXR-LBD in presence of an agonistic ligand entirely sets the co-repressor free (Fig. 5b, e). Notably, no signal of the co-activator Cβ was visible. For the synthetic FXR agonists GW4064 and fexaramine, equal results were obtained (Fig. 5f, g). For all three FXR agonists, integration of co-activator and co-repressor signals in the ternary mixtures showed a clear preference for co-activator binding indicated by high free co-repressor/free co-activator ratios (CDCA: 87/13; GW4064: 86/14; fexaramine: 77/23).

Next, we applied the FXR antagonist guggulsterone[31] to the same NMR experiments (Supplementary Fig. 7). The $^1$H-$^{15}$N-HSQC of the quaternary mixture of FXR-LBD, guggulsterone, co-activator and co-repressor peptide, revealed no co-repressor signals while the Cβ signals of the co-activator were fully present without detectable shifts. This observation confirmed the

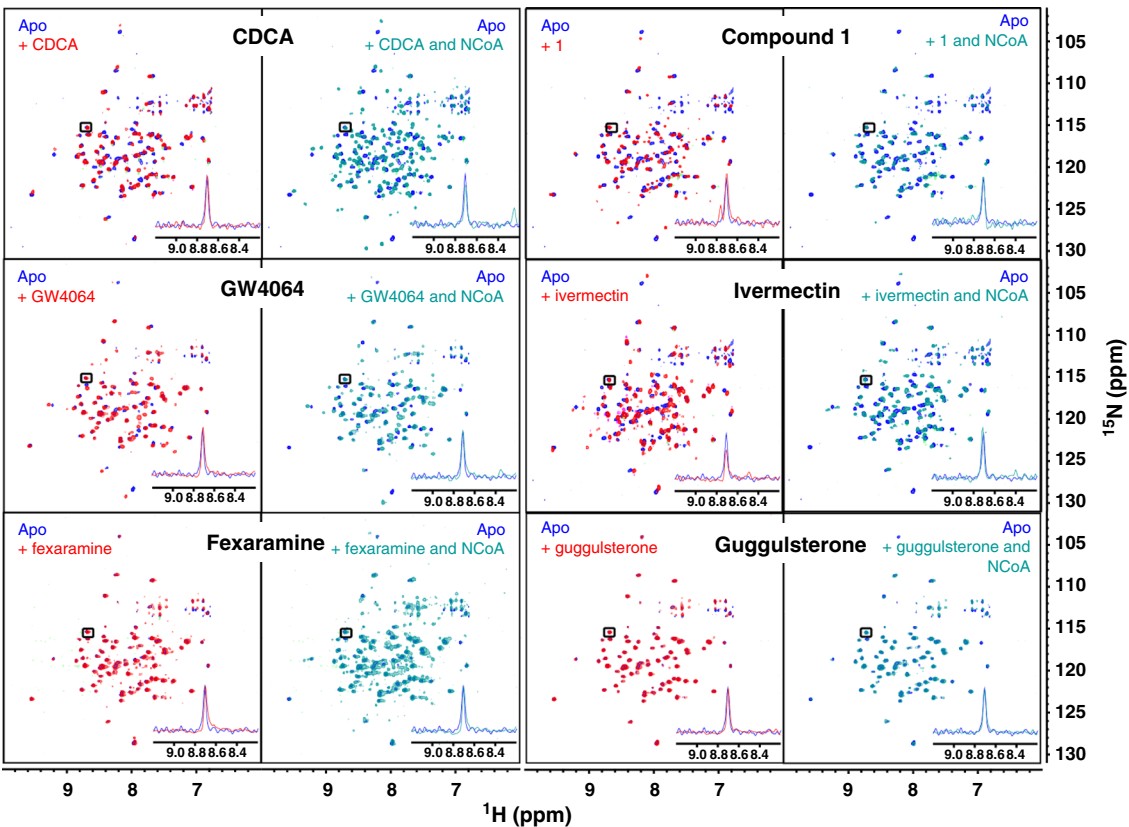

**Fig. 4** $^1$H/$^{15}$N-HSQC spectra of fully labeled FXR-LBD. Left panels show superposition of apo and apo+ligand, right panels show superposition of apo and apo + ligand + co-activator peptide. Blue—apo; red—with ligand; cyan—with ligand and co-activator peptide (NCoA). Concentrations of protein (200 µM), peptide (500 µM) and ligand (500 µM) were constant for all experiments to ensure saturation. All spectra were processed and compared at the same S/N. A representative signal is shown in each panel to visualise equal S/N. In unliganded state, the $^1$H/$^{15}$N-HSQC reveals only a fraction of all signals indicating conformational flexibility of part of the FXR-LBD. Upon addition of an FXR agonist (CDCA, GW4064 or fexaramine), additional but still not all signals appear suggesting partial stabilisation of the protein. Addition of co-activator peptide causes marked further stabilisation as observed by appearance of almost all $^1$H/$^{15}$N-HSQC signals. Addition of the FXR partial agonists ivermectin or **1** also induced stabilisation of the FXR-LBD but in contrast to the FXR agonists, addition of co-activator peptide had no further effect. Addition of the FXR antagonist guggulsterone had only minor/no effect on the HSQC indicating much less stabilisation of the FXR-LBD

assumption that the FXR-LBD in presence of an antagonist is fully complexed with co-repressor and that the co-activator is not bound.

According to our NMR experiments, unliganded FXR-LBD in solution recruits the co-repressor peptide, which is partly released upon agonist binding. The equilibrium of bound and free co-repressor in presence of a full agonistic ligand but in absence of co-activator slightly differs for distinct agonists but in all cases shifts to strong release of the co-repressor upon addition of co-activator peptide. FXR antagonist guggulsterone, in contrast, stabilised the interaction of the FXR-LBD with the co-repressor which is not released even in presence of co-activator. Hence, the NMR experiments allow a clear differentiation of agonistic and antagonistic ligands.

Knowing these effects on co-repressor and co-activator binding caused by FXR agonists and antagonists we then analysed the behaviour of partial agonists ivermectin and **1** in this setting (Fig. 5h, i). Their addition to the FXR-LBD-co-repressor complex led to slight reappearance of the Cβ signals of NCoR indicating that the partial agonists partly displace the co-repressor from the FXR-LBD. When the NCoA peptide was added to this system, however, the Cβ signals of both the co-repressor and the co-activator were broadened but visible. Free co-repressor/free co-activator ratios of 30/70 and 25/75 observed for the partial

agonists **1** and ivermectin, respectively, clearly indicate that a fraction of both peptides is bound to the FXR-LBD protein. Moreover, the Cβ signals of co-activator and co-repressor revealed significant chemical shift perturbations (CSP) and line broadening which confirms the binding of both peptides. Thus, binding of partial agonists to the FXR-LBD induces a conformation that is capable of binding co-activator and co-repressor. This FXR-LBD conformation having affinity to co-activator as well as co-repressor explains the pharmacological behaviour of partial FXR agonists and, altogether, this indicates that FXR activity is not either switched on or off but can be tuned to several stages in between.

In addition to explaining the structural and mechanistic basis of partial FXR activation, our results suggest that not only ligand binding affects the conformation of the FXR-LBD but that co-activator binding has even stronger influence. According to current understanding, agonist binding to FXR causes conformational changes in the FXR-LBD that lead to stabilisation of α12 at the core domain which is a mandatory prerequisite to co-activator binding. However, this would indicate that co-activator binding has less stabilising effects on the FXR-LBD than ligand binding. To resolve this anthology, we studied the conformation and position of α12 in the unliganded FXR-LBD to figure out whether it follows the canonical model for NRs of a mouse trap.

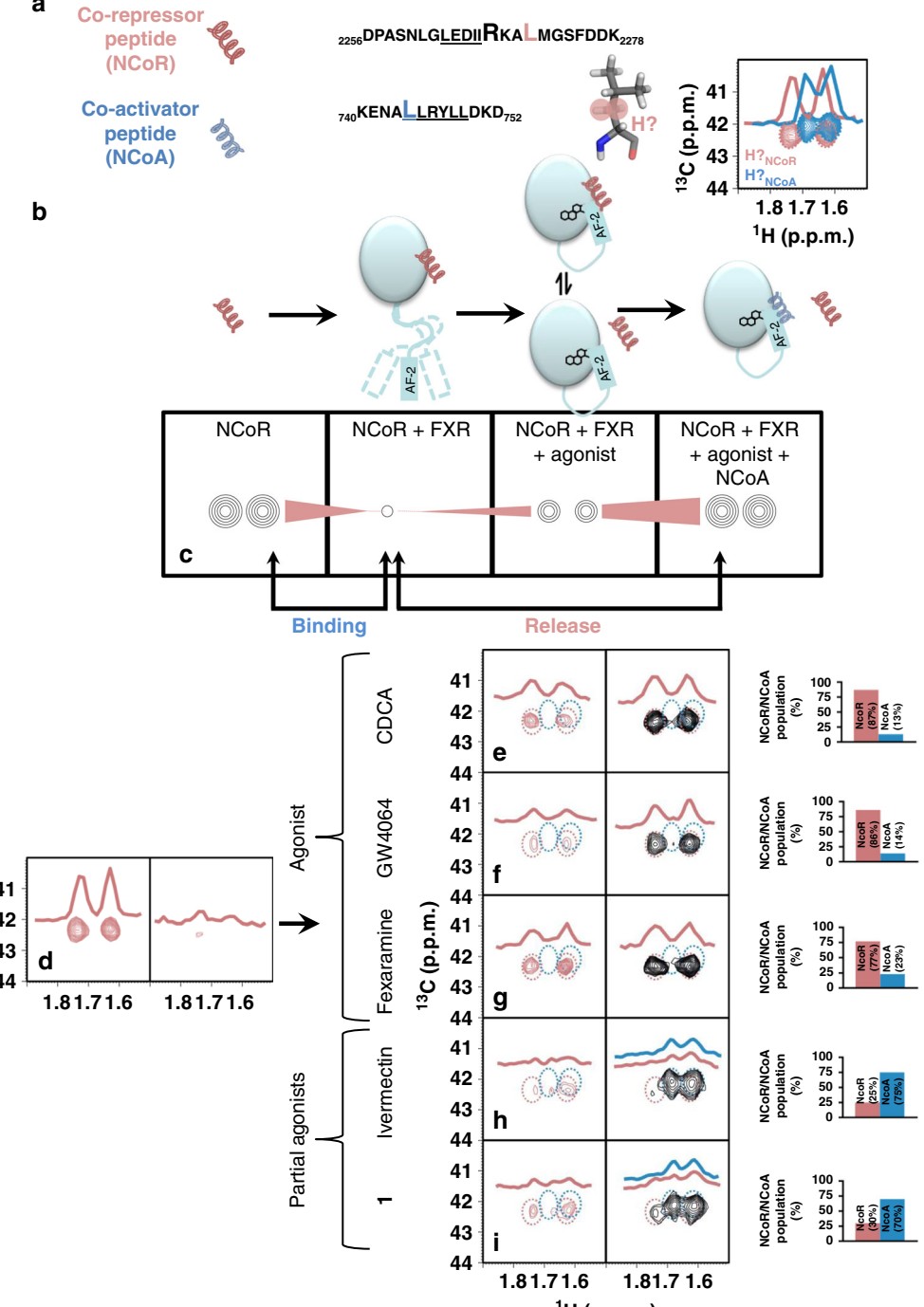

**Fig. 5** NMR studies on the FXR activation mechanism. **a** Amino acid sequence of co-repressor (NCoR) and the co-activator (NCoA) with $^{15}$N/$^{13}$C-isotpically enriched amino acids highlighted with larger font. A stick model of leucine with its Hβ highlighted. The inset shows the Hβ-region of $^{1}$H, $^{13}$C-HSQC spectra of the labelled peptides. The chemical shifts for the NCoA and NCoR are not overlapping and are easily distinguishable. **b** Schematic representation of the typical behaviour of an LBD in response to ligand-dependent activation and **c** resulting changes in the NMR signal (Hβ of leucine) of the co-repressor (NCoR) at different points of interaction. Disappearance of NMR signals upon binding to FXR, partial reappearing (release) upon addition of agonist and further reappearance upon addition of NCoA indicating complete release of NCoR. **d** Monitoring of the Hβ-signal of leucine in NCoR in response to addition of FXR, followed by addition of ligand (**e–i**) and the co-activator peptide: The Hβ signals of the co-repressor (**d**) are severely broadened upon addition of FXR, indicating binding. Upon addition of an agonists (**e–g**), an increase in signal intensity is observed indicating that agonist binding induces partial release of NCoR (**e–g**, left spectra). Further, upon addition of the co-activator peptide, the intensity of the NCoR signals further increases and the signals of NCoA disappear (blue circled region), indicating the complete release of the co-repressor and binding of NCoA to the FXR-agonist complex (**e–g**, right spectrum). The bar graphs adjacent to the spectra represent the relative populations of the released NCoR/NCoA peptides upon binding of the ligand. Populations of each reporter signal were determined from their intensities as P[(CDCA$^{NCoR}$) = 100•I$^{NCoR}$/(I$^{NCoR}$ + I$^{NCoR}$)] and vice versa. The spectra obtained in the presence of partial agonists (**h**, **i**) show marginal increase in the intensity of the repressor signals and minor line broadening for the signals of the activator, indicating that both co-repressor and the co-activator bind. This is also reflected by the relative populations of the released NCoR/NCoA peptides. Antagonist guggulsterone, in contrast, in this setting induces full binding of NCoR peptide with no recruitment of NCoA peptide (see Supplementary Fig. 7)

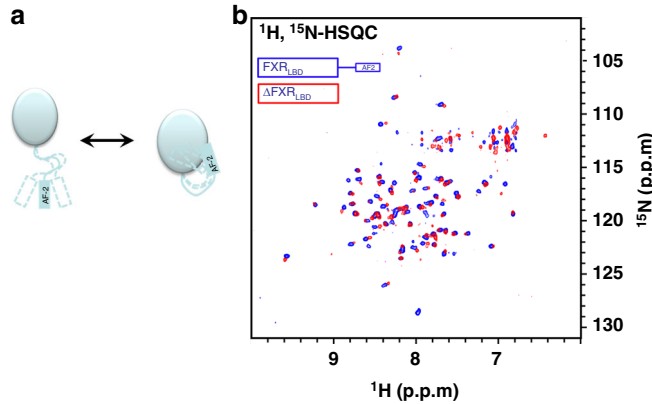

**Fig. 6** NMR study on the interaction of α12 with the FXR-LBD core domain. **a** Model depicting transient interaction of the AF2 helix with the core domain. **b** Superimposition of ¹H, ¹⁵N-HSQC of fully 13C/15N-labeled FXR-LBD (blue) and an AF2 truncated FXR-LBD mutant (red) reveals significant shifts and disappearance of signals indicating that α12 interacts transiently with the core domain of the FXR-LBD even in the absence of a ligand/agonist

For this purpose, we recorded ¹H/¹⁵N-HSQC spectra of ¹³C/¹⁵N-labeled FXR-LBD and a ¹³C/¹⁵N-labeled FXR-LBD mutant lacking α12 (Fig. 6). Considering the commonly assumed mechanism that involves recruitment of α12 upon agonist binding, both spectra should hardly differ in absence of a ligand since no signals should be present for destabilised α12 in solution. Superimposition of both spectra, however, revealed significant differences with additional signals and marked CSP. This indicates that the activation function α12 in case of FXR is not disordered and exposed to solvent but even in absence of a ligand is partially stabilised and part of the core LBD. In addition, it may suggest that there are conformational changes in presence of α12 affecting the whole region involving α11 and α12 as well as the connecting loop, which both agrees well with our observations in this study.

## Discussion

The nuclear receptor FXR has experienced considerable interest in drug discovery and pharmacology in recent past due to its important role in metabolism and its value as drug target to treat liver disorders and metabolic diseases. With OCA, the first FXR targeting agent has already received market approval and is expected to gain expanded license for further indications such as NASH. Hence, the steroidal drug has validated FXR as promising target and confirmed its therapeutic value. Still, due to its nature as NR participating in various endocrine functions, FXR modulation is also prone to cause mechanism-based side-effects. Clinical trials of OCA have already shown that extensive FXR activation disrupts cholesterol homoeostasis with FXR activation blocking metabolic conversion of cholesterol to bile acids via SHP/FGF19 up- and CYP7A1 down-regulation. Since this pathway constitutes the main route of metabolic cholesterol elimination, its long-term pharmacological blockade may have serious consequences.

Partial activation appears to be an avenue to safely exploit FXR as drug target as it could reduce side effects such as loss of metabolic cholesterol degradation. Similar strategies have been proposed for other NRs that are known for causing unwanted effects such as PPARs. However, to explore this strategy, a better understanding of partial FXR activation on molecular level is necessary.

Therefore, we have studied the structural determinants of partial FXR agonism by solving co-crystal structures and investigating the conformational dynamics of quaternary complexes involving the FXR-LBD, co-activator and co-repressor peptides as well as a variety of FXR ligands by NMR spectroscopy. We have used the potent partial FXR agonist **1** which caused partial induction of FXR target genes compared to CDCA in hepatocytes and in vivo confirming its partial agonistic properties.

Initially, we solved the co-crystal structures of partial agonist **1** and the endogenous agonist **2** in complex with the human FXR-LBD which was previously not available. The structural information obtained from the comparison of FXR-LBD co-crystal structures in complex with agonists, antagonists and partial agonist **1** revealed several differences. Overall, the sum of minor differences seems to affect the conformation of the FXR-LBD explaining different pharmacological effects. The only remarkable difference is found in a shifted binding mode formed by **1** compared to full agonists. By occupying additional space towards the loop connecting helices α11 and α12 in the ligand binding pocket, the partial agonist causes an outward movement of W454 and destabilises the loop connecting α12 with the core LBD. As a consequence, the positions of α12 and the co-activator bound to the FXR-LBD are shifted and the interaction between co-activator and FXR-LBD is weakened by losing five H-bonds as well as several non-bonded contacts. The co-crystal structures, therefore, indicated that the mechanistic difference in agonist and partial agonist is explained by conformations of the FXR-LBD having unequal affinity to the co-activator and vice versa. However, a co-crystal structure represents only a single snapshot of a dynamic binding equilibrium and was in this case not sufficient to gain mechanistic understanding. This is further supported by the discrepancy between the activity of FXR modulator ivermectin behaving as partial agonist in vitro and the FXR-LBD complex containing ivermectin that represents an inactive state of the NR as it is bound to the nuclear co-repressor NCoR-1. Moreover, analysis of all NR structures deposited in the PDB revealed that binding of the co-activator itself strongly affects the conformation of NR ligand binding domains tampering structural data. To avoid these effects, experiments in solution were necessary. Therefore, we studied the structure of the FXR-LBD as well as its interaction with co-activator and co-repressor in solution by NMR.

¹H/¹⁵N-HSQC of fully ¹⁵N-labeled FXR-LBD revealed that agonist binding though stabilising the protein causes less conformational stability than co-activator binding, which induced remarkable appearance of additional signals. Furthermore, we observed that the co-activator robustly stabilised the FXR-LBD in presence of any agonist but failed to do so with the partial agonists **1** and ivermectin further confirming that the functional conformation of all agonistic structures is similar but significantly differs from a partial agonistic conformation. Notably, the effect of the partial agonists on the FXR-LBD in solution merely resembled the effect of an antagonist.

Going more into detail we subsequently focused on labeled co-activator and repressor in NMR and studied their behaviour upon agonist, partial agonist or antagonist binding to the FXR-LBD. Unliganded FXR-LBD robustly recruited the co-repressor which could be observed by disappearance of co-repressor signals upon addition of the LBD. While agonists induced significant displacement of the co-repressor, full release only happened in presence of the co-activator peptide leading to entire reappearance of the co-repressor Cβ signals while no Cβ signal of the co-activator was present. Congruently, antagonist guggulsterone caused the opposite by stabilising the interaction of LBD and co-repressor which was hardly released even in presence of the co-activator. In this case, only the Cβ signal of the co-activator was

present in the NMR spectra. Studying the partial agonists **1** and ivermectin in NMR revealed moderate release of the co-repressor. However, in presence of the co-activator peptide, both co-regulators bound to the FXR-LBD as observed by presence of Cβ signals of co-activator as well as co-repressor together with line-broadening of all Cβ signals. When we quantified the ratios between free co-repressor to free co-activator in solution upon addition of the diverse FXR ligands to the FXR-LBD, we could again clearly distinguish between agonists and antagonists. For the FXR activators, we observed high free co-repressor / free co-activator ratios (77:23–87:13) indicating strong preference for co-activator binding. FXR antagonist guggulsterone, in contrast, fully stabilised the complex of the FXR-LBD with the co-repressor and no co-activator was bound to the FXR-LBD (ratio 15:85). In accordance with the other experiments in solution, partial agonists **1** and ivermectin adopted a state between agonists and antagonist displaying free co-repressor/free co-activator ratios of 30:70 and 25:75, respectively.

The results from co-crystal structure investigations and NMR experiments in solution together draw a sound mechanism for FXR activation that also explains partial agonism: The unliganded FXR-LBD is bound to co-repressor in solution. An agonistic ligand destabilises the interaction of FXR-LBD and co-repressor which leads to partial dissociation of the co-repressor from the NR. In presence of a co-activator, the co-repressor is then completely released, and the co-activator is bound to the FXR-LBD. An antagonist, in contrast, stabilises the interaction between FXR-LBD and co-repressor. Therefore, the co-repressor is not even released in presence of a co-activator, which can consequently not bind to the NR. A partial agonist behaves like an agonist by weakening the FXR-LBD-co-repressor interaction but in contrast to a full agonist it induces a conformation of the FXR-LBD that has affinity to the co-activator and the co-repressor. This is congruent with our observation that outward movement of W454 and destabilisation of the AF-2 loop in the co-crystal structure of the FXR-LBD in complex with partial agonist **1** causes a shifted position of α12, which in turn affects position and strength of binding of the co-activator.

Structurally, unliganded FXR-LBD comprises the α11 and L: α3-α4 regions in an unordered state and can bind the co-repressor (Fig. 7a, b). An FXR agonist induces α11-formation from the unordered region between α10 and α12 and promotes formation of a 3$_{10}$ helix in the loop region between α3 and α4 (L: α3-α4) thereby stabilising the FXR-LBD and promoting co-activator binding (Fig. 7d–g). In antagonistic conformation (Fig. 7c), the α11 and α12 regions of the FXR-LBD are unordered, the α3–α4 region exists as a loop and the Ω-loop (L:α2′-α3) is destabilised suggesting that antagonist binding does not cause marked FXR-LBD stabilisation. The partial agonist **1** induces a conformation of the FXR-LBD that has agonistic and antagonistic elements (Fig. 7h–j). **1** promotes α11 formation as FXR agonists but to a lesser extent and does not stabilise the α3-α4 loop in a helical state. Thereby, the partial agonist enables an FXR-LBD conformation that has affinity to co-activators and co-repressors.

Mutagenesis of W454 in the FXR-LBD confirmed crucial involvement of this residue in FXR modulation. FXR mutants W454A and W454Y strongly differed in their responsiveness to FXR ligands. While the FXR agonist GW4064 activated wt-FXR and both mutants with comparable efficacy and potency, the endogenous FXR agonist CDCA failed to activate W454A and W454Y. These observations agree with the FXR-LBD co-crystal structures in complex with GW4064 (3DCT) and CDCA (6HL1), which suggest involvement of Trp454 in binding of CDCA but not GW4064 (Supplementary Fig. 8). Phe284 and Leu287 are involved in the binding of the isopropyl moiety of GW4064 while the opposing Trp454 is distant from the hammerhead structure.

CDCA, in contrast, is not stabilised by Phe284/Leu287 but the hydrophobic β-face of its A-ring makes a lipophilic contact to Trp454 which is crucial to bring the 3α-hydroxyl group in a suitable position to interact with Tyr361 and His447. Taken together, Trp454 plays an important role in mediating the interaction of FXR with its ligands. The FXR modulator ivermectin revealed enhanced activation efficacy on W454Y but was inactive on W454A and partial agonist **1** markedly gained in potency and activation efficacy on both mutants compared to wt-FXR. Together, these observations suggest that the tryptophan residue, as the most rare and lipophilic amino acid, constitutes a key residue to mediate FXR modulation. By kicking out W454 from its inward facing position, partial agonist **1** causes specific modulatory effects on FXR activity that clearly differ from classical FXR agonists.

In addition to deciphering the molecular mode of partial FXR activation, our data reveal that an agonist as a prerequisite of FXR activation induces a conformation of the LBD that favours co-activator binding but is still partly capable of co-repressor binding. Only binding of the co-activator peptide finally releases the co-repressor and stabilises the active conformation of FXR. Consequently, which kind of co-regulator is present and binds to FXR ultimately governs whether the NR is activated or not. In accordance with this, our observation of significantly different spectra for fully $^{13}$C/$^{15}$N-labeled FXR-LBD and the labeled truncated mutant lacking α12 shows that AF2 is already part of the core domain in absence of an agonist and not unordered and exposed to solvent. Therefore, FXR does not follow the canonical mouse trap mechanism of NR activation.

Our observations on the molecular mechanism of FXR activation are in accordance with previous studies[14] on FXR and other NRs suggesting that the FXR-LBD can adopt more conformations than an active and an inactive state. In contrast to the initially proposed mouse trap mechanism and later refined models suggesting an unordered conformation of α12 in solution in absence of an agonist[14], our results indicate that α12 is ordered and forms part of the core LBD also in absence of a ligand. Moreover, the two FXR-LBD apo-structures 5Q0K[32] and 6HL0 lack a ligand but are still complexed with a co-activator peptide suggesting that also unliganded FXR is able to recruit co-activators. This agrees with our observation of α12 being part of the core LBD also in apo-state as it is required for co-activator binding. However, the recruitment of co-activator by the apo-form is not reflected by our NMR studies and might also be a crystallisation artefact. Still, it is possible that the apo-state has a very weak affinity to co-activators that is too low to be captured in our system. Beyond explaining the molecular basis of partial FXR activation, our results imply that interaction of FXR with different co-regulator proteins including distinct co-activators and co-repressors can possibly be controlled through ligand design. Since ligand binding governs the conformation adopted by the NR's ligand binding domain and this conformation in turn affects the affinity of the complex to co-regulators, development of co-regulator specific FXR modulators seems possible. Thereby, tissue- or even gene-selective FXR modulation may be achievable.

Recapitulating, we report the structure of human FXR in complex with the natural agonist CDCA and have developed a sound mechanism for partial FXR activation. This better understanding of FXR's molecular mechanism may significantly support future drug discovery targeting the NR and aid the development of safer FXR modulating agents.

## Methods
**Chemistry.** General: Reactions were carried out under argon atmosphere in dried glassware. Reagents and solvents were purchased from commercial providers and used without further purification.

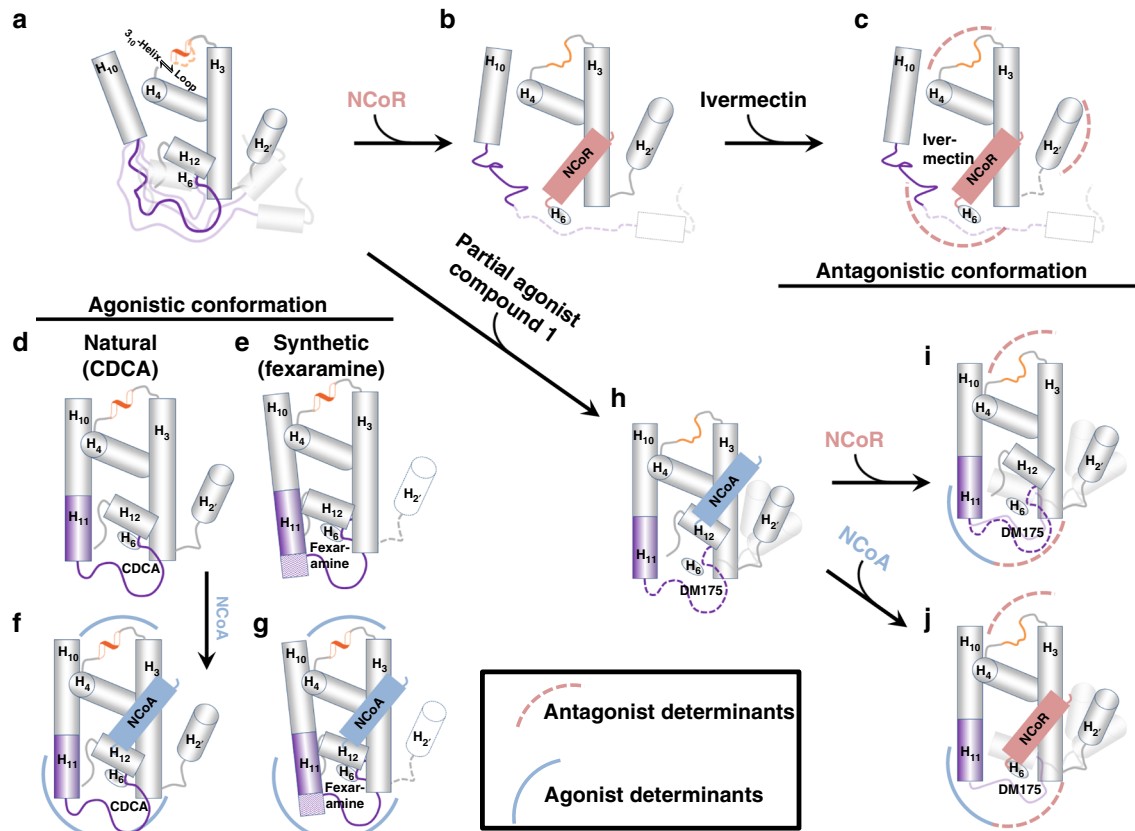

**Fig. 7** Proposed model for partial agonism. Regions in the FXR-LBD structures with major changes induced upon binding of ligands (agonist, antagonist and partial agonist) are highlighted with arcs (red/dashed for antagonist characteristics, blue/bold for agonist characteristics). **a** In the apo FXR, the α11 adopts a loop structure and the α3-α4 region predominantly exists as a loop. **b** Co-repressor (NCoR) binds to the apo FXR. **c** Antagonist binding does not induce significant structural changes, however, destabilises the α11-α12 (AF2-helix). The α3-α4 loop adopts a loop structure and the Ω-loop (α2'-α3 loop region) is destabilised. **d** Binding of the natural agonist CDCA, induces partial formation of α11 (purple) and also the α3-α4 loop adopts a 3₁₀—helix conformation. **e** Binding of the synthetic agonist fexaramine, induces extended α11 formation (purple) and the α3-α4 loop adopts a 3₁₀— helix conformation. **f** Co-activator (NCoA) binds to the CDCA bound FXR-LBD. **g** Co-activator (NCoA) binds to the FXR-LBD bound to fexaramine. **h** Binding of the partial agonist **1**, induces α11 formation (purple), destabilises the α11-α12 (AF2) and the Ω-loop region. The α3-α4 loop adopts a loop conformation as observed for the antagonist binding. **i**, **j** Partial agonist **1** bound FXR is competent of binding either the co-activator (NCoA) or the co-repressor (NCoR)

4-(2-Aminobenzamido)benzoic acid: 4-Aminobenzoic acid (1.2 g, 9.0 mmol, 1.5 eq) was dissolved in ethanol (abs., 30 mL) and heated to reflux. Then, isatoic anhydride (0.98 g, 6.0 mmol, 1.0 eq) was added to the solution with the immediate formation of $CO_2$. After the evolution of $CO_2$ was finished and a clear brown solution had formed (~60 min), the mixture was cooled to room temperature and filtered. The solvent was then partly evaporated under reduced pressure until crystallisation of the product started. The crude product was filtered off and recrystallised twice from hexane/ethyl acetate/acetic acid to obtain the title compound as colourless solid (1.2 g, 75 %). ¹H NMR (500 MHz, DMSO-$d_6$) δ = 10.62 (s, 1 H), 7.95−7.91 (m, 2 H), 7.91–7.86 (m, 2 H), 7.80 (d, $J$ = 7.6 Hz, 1 H), 7.42 (t, $J$ = 7.3 Hz, 1 H), 7.14 (d, $J$ = 7.2 Hz, 1 H), 7.03 (d, $J$ = 5.8 Hz, 1 H) ppm. ¹³C NMR (126 MHz, DMSO-$d_6$) δ = 167.42, 167.30, 143.70, 132.94, 132.46, 130.63, 129.73, 129.64, 126.24, 125.96, 120.35, 120.10 ppm. $C_{14}H_{12}N_2O_3$. MS (ESI-): m/z 255.0 ([M-H]⁻, 100).

4-(2-(4-*tert*-Butylbenzamido)benzamido)benzoic acid (**1**): 4-(2-Aminobenzamido)benzoic acid (1.0 g, 4.0 mmol, 1.0 eq) was dissolved in THF (abs., 20 mL) and pyridine (abs., 4.0 mL) was added. After a clear solution had formed, 4-*tert*-butylbenzoyl chloride (1.0 g, 1.00 mL, 5.2 mmol, 1.3 eq) was added drop-wise. The reaction mixture was kept stirring at room temperature and the reaction progress was monitored by TLC. After 4 h, the reaction mixture was diluted with ethyl acetate, washed three times with 10% hydrochloric acid and dried over $Na_2SO_4$. Further purification was performed by column chromatography on silica using hexane/ethyl acetate with 2% acetic acid as mobile phase to obtain the title compound as colourless solid (1.1 g, 67 %). ¹H NMR (500 MHz, DMSO-$d_6$) δ = 12.80 (s$_{br}$, 1 H), 11.43 (s, 1 H), 10.79 (s, 1 H), 8.42 (dd, $J$ = 8.3, 0.8 Hz, 1 H), 7.97–7.94 (dt, $J$ = 8.7, 1.9 Hz, 2 H), 7.92 (dd, $J$ = 7.9, 1.5 Hz, 1 H), 7.87 (dt, $J$ = 6.7, 2.0 Hz, 2 H), 7.85 (dt, $J$ = 6.7, 2.1 Hz, 2 H), 7.63 (td, $J$ = 7.8, 1.4 Hz, 1 H), 7.59 (dt, $J$ = 8.6, 1.9 Hz, 2 H), 7.31 (td, $J$ = 7.7, 1.1 Hz, 1 H), 1.31 (s, 9 H) ppm. ¹³C NMR (126 MHz, DMSO-$d_6$) δ = 168.17, 167.39, 165.11, 155.46, 143.25, 139.03, 132.85, 132.20, 130.67, 129.62, 127.43, 126.43, 126.18, 123.77, 123.68, 122.02, 120.57, 35.21, 31.34 ppm. $C_{25}H_{24}N_2O_4$. MS (ESI-): m/z 415.9 ([M − H]⁻, 100);

HRMS (ESI+): calculated for $C_{25}H_{24}N_2O_4Na$: m/z 439.16283, found: m/z 439.16256 ([M + Na]⁺). Combustion analysis: measured (calculated): C 71.72 (72.10), H 5.78 (5.81), N 6.77 (6.73). ¹H- and ¹³C-NMR of **1** are shown in Supplementary Fig. 9.

**In vitro pharmacology**. FXR transactivation assay: Plasmids: pcDNA3-hFXR[33] contains the sequence of human FXR. pSG5-hRXRα[34] contains the sequence of human RXRα. pGL3basic (Promega Corporation, Fitchburg, WI, USA) was used as a reporter plasmid and contains a shortened construct of the promotor of the bile salt export protein (BSEP) cloned into the SacI/NheI cleavage site in front of the luciferase gene[35]. pRL-SV40 (Promega) served as a control for normalisation of transfection efficiency and cell growth. Assay procedure: HeLa cells (Deutsche Sammlung von Mikroorganismen und Zellkulturen GmbH (DSMZ), Leibniz-Institut, Braunschweig, Germany) were grown in DMEM high glucose supplemented with 10% fetal calf serum (FCS), sodium pyruvate (1 mM), penicillin (100 U/mL) and streptomycin (100 μg/mL) at 37 °C and 5% $CO_2$. 24 h before transfection, HeLa cells were seeded in 96-well plates with a density of 8000 cells per well. 3.5 h before transfection, medium was changed to DMEM high glucose, supplemented with sodium pyruvate (1 mM), penicillin (100 U/mL), streptomycin (100 μg/mL) and 0.5% charcoal-stripped FCS. Transient transfection of HeLa cells with BSEP-pGL3, pRL-SV40 and the expression plasmids pcDNA3-hFXR and pSG5-hRXR was carried out using calcium phosphate transfection method. Sixteen hours after transfection, medium was changed to DMEM high glucose, supplemented with sodium pyruvate (1 mM), penicillin (100 U/mL), streptomycin (100 μg/mL) and 0.5% charcoal-stripped FCS. 24 h after transfection, medium was changed to DMEM without phenol red, supplemented with sodium pyruvate (1 mM), penicillin (100 U/mL), streptomycin (100 μg/mL), L-glutamine (2 mM) and 0.5% charcoal-stripped FCS, now additionally containing 0.1% DMSO and the respective test compound or 0.1% DMSO alone as untreated control. Each sample/concentration was tested in triplicate wells and each experiment was repeated

independently at least three times. Following 24 h incubation with the test compounds, cells were assayed for luciferase activity using Dual-Glo™ Luciferase Assay System (Promega) according to the manufacturer's protocol. Luminescence was measured with a Tecan Infinite M200 luminometer (Tecan Deutschland GmbH, Crailsheim, Germany). Normalisation of transfection efficiency and cell growth was done by division of firefly luciferase data by renilla luciferase data multiplied by 1000 resulting in relative light units (RLU). Fold activation was obtained by dividing the mean RLU of the tested compound at a respective concentration by the mean RLU of untreated control. Relative activation was obtained by dividing the fold activation of the tested compound at a respective concentration by the fold activation of FXR full agonist GW4064 at 3 µM. $EC_{50}$ and standard deviation values were calculated with the mean relative activation values of at least three independent experiments set up in triplicates by SigmaPlot 12.5 (Systat Software GmbH, Erkrath, Germany) using a four parameter logistic regression. The assay was validated with FXR agonists OCA ($EC_{50} = 0.16 \pm 0.02$ µM, $87 \pm 3\%$ rel. max. act.) and GW4064 ($EC_{50} = 0.51 \pm 0.16$ µM, 3 µM defined as 100%)[24,36]. FXR mutants were prepared by designing an appropriate primer with the single amino acid mutation and following the standard QuickChange® site-directed mutagenesis protocol. Assay procedure was performed as described for the wild type.

FXR target gene quantification (qRT-PCR): Cell culture: HepG2 cells (DSMZ) were grown in DMEM high glucose, supplemented with 10% fetal calf serum (FCS), sodium pyruvate (SP, 1 mM), penicillin (100 U/mL) and streptomycin (100 µg/mL) at 37 °C and 5% $CO_2$. HT29 cells were grown in McCoy's 5 A medium, supplemented with 10% FCS, penicillin (100 U/mL) and streptomycin (100 µg/mL) at 37 °C and 5% $CO_2$. Cells were seeded in six-well plates ($2 \times 10^6$ per well) in the same medium. Twenty four after seeding, medium was changed to MEM supplemented with 1% charcoal-stripped FCS (CS-FCS), penicillin (100 U/mL), streptomycin (100 µg/mL) and L-glutamine (2 mM). After additional 24 h, medium was again changed now additionally containing the test compounds in DMSO or DMSO alone (final concentration 0.1% DMSO). Cells were incubated with the test compounds for 6, 18 or 24 h, harvested, washed with cold PBS and then directly used for RNA extraction. RNA extraction and cDNA synthesis: Two micrograms of total RNA extracted from above described cells with the E. Z. A. Total RNA Mini Kit (R6834-02, Omega Bio-tek Inc., Norcross, GA, U.S.A.) were reversely transcribed into cDNA using the High-Capacity cDNA Reverse Transcription Kit (4368814, Life Technologies, Carlsbad, CA, U.S.A.) according to the manufacturer's protocol. q-RT PCR: FXR target gene expression was evaluated by quantitative PCR analysis with a StepOnePlus™ System (Life Technologies) using PowerSYBRGreen (Life Technologies; 12.5 µL per well) and the primers (300 nM each) listed in Supplementary Table 2. Results were normalised to GAPDH Ct values. Each sample was set up in triplicates and repeated in at least three independent experiments. The expression was quantified by the comparative ΔΔCt method. Results (expressed as mean ± SEM% mRNA expression compared to DMSO (0.1%); $n \geq 3$): BSEP: CDCA (50 µM): $624 \pm 33\%$; 1 (5 µM): $205 \pm 2\%$. SHP: CDCA (50 µM): $537 \pm 98\%$; 1 (5 µM): $262 \pm 83\%$. CYP7A1: CDCA (50 µM): $33 \pm 7\%$; 1 (5 µM): $61 \pm 3\%$. OSTα: CDCA (50 µM): $813 \pm 159\%$; 1 (5 µM): $297 \pm 34\%$. IBABP: CDCA (50 µM): $306 \pm 17\%$; 1 (5 µM): $137 \pm 10\%$. FGF19: CDCA (50 µM): $221 \pm 18\%$; 1 (5 µM): $164 \pm 8\%$.

Hybrid reporter gene assays: Plasmids: The Gal4-fusion receptor plasmids pFA-CMV-hPPARα-LBD[37], pFA-CMV-hPPARγ-LBD[37], pFA-CMV-hPPARδ-LBD[37], pFA-CMV-hLXRα-LBD[38], pFA-CMV-hLXRβ-LBD[38], pFA-CMV-hRXRα-LBD[39], pFA-CMV-hRXRβ-LBD[39], pFA-CMV-hRXRγ-LBD[39], pFA-CMV-hRARα-LBD[39], pFA-CMV-hRARβ-LBD[39], pFA-CMV-hRARγ-LBD[39], pFA-CMV-hVDR-LBD[39], pFA-CMV-hCAR-LBD[39] and pFA-CMV-hPXR-LBD[39] coding for the hinge region and ligand binding domain (LBD) of the canonical isoform of the respective nuclear receptor have been reported previously. pFR-Luc (Stratagene, San Diego, CA, U.S.A.) was used as reporter plasmid and pRL-SV40 (Promega) for normalisation of transfection efficiency and cell growth. Assay procedure: HEK293T cells (DSMZ) were grown in DMEM high glucose, supplemented with 10% FCS, sodium pyruvate (1 mM), penicillin (100 U/mL) and streptomycin (100 µg/mL) at 37 °C and 5% $CO_2$. The day before transfection, HEK293T cells were seeded in 96-well plates ($2.5 \times 10^4$ cells/well). Before transfection, medium was changed to Opti-MEM without supplements. Transient transfection was carried out using Lipofectamine LTX reagent (Invitrogen, Carlsbad, CA, U.S.A.) according to the manufacturer's protocol with pFR-Luc, pRL-SV40 (Promega) and the respective pFA-CMV-hNR-LBD. 5 h after transfection, medium was changed to Opti-MEM supplemented with penicillin (100 U/mL), streptomycin (100 µg/mL), now additionally containing 0.1% DMSO and the respective test compound or 0.1% DMSO alone as untreated control. Each concentration was tested in duplicates and each experiment was repeated independently three times. Following overnight (12–14 h) incubation with the test compounds, cells were assayed for luciferase activity using Dual-Glo™ Luciferase Assay System (Promega) according to the manufacturer's protocol. Luminescence was measured with an Infinite M200 luminometer (Tecan) or a Tecan Spark 10 M luminometer (Tecan). Normalisation of transfection efficiency and cell growth was done by division of firefly luciferase data by renilla luciferase data and multiplying the value by 1000 resulting in relative light units (RLU). Fold activation was obtained by dividing the mean RLU of a test compound at a respective concentration by the mean RLU of untreated control (0.1% DMSO). Reference agonists (PPARα: GW7647; PPARγ: pioglitazone; PPARδ: L165,041; LXRα/β: T0901317; RXRs: bexarotene; RARs: tretinoin; VDR: calcitriol; CAR: CITCO; PXR: SR12813) were included in every individual assay as

positive control to monitor performance and all hybrid assays were validated with the above-mentioned reference agonists which yielded $EC_{50}$ values in agreement with literature.

TGR5 assay: The activity of 1 on TGR5 was evaluated by measuring the level of cAMP using a HTR-FRET assay. In brief, NCI-H716 cells were cultured in DMEM supplemented with 10%FBS, using 96-well plates coated with Matrigel (BD Biosciences, Franklin Lakes, NJ, U.S.A.). After 24 h, cells were stimulated with test compound 1 or lithocholic acid as positive control for 60 min at 37 °C in OptiMEM with 1 mM IBMX (Sigma Aldrich, St. Louis, MO, U.S.A.). The level of intracellular cAMP was assessed using the Lance kit (Perkin Elmer, Waltham, MA, U.S.A.) according to the manufacturer's protocol. Each experiment was repeated independently three times in duplicates.

Cytotoxicity assay: LDH assay (Roche Diagnostics, Rotkreuz, Switzerland) was performed according to manufacturer's instructions. In brief, HepG2 cells (DSMZ) were seeded in DMEM containing 1% SP, 1% PS and 10% FCS in 96-well plates ($3 \times 10^4$ cells/well). After 24 h medium was changed to DMEM containing penicillin (100 U/mL), streptomycin (100 µg/mL) and 1% FCS, and cells were incubated with the respective compounds for 48 h. As positive control, TRITON X-100 (2%) was added 1 h before measurement. After incubation, supernatant of each well was transferred into a fresh plate and LDH substrate/reagent was added. After 20 min incubation absorption at measurement (490 nm) and reference (690 nm) wavelength was determined with a TECAN infinite 200 (Tecan). All experiments were performed in triplicates in four independent repeats. Results (expressed as mean ± SEM LDH-activity; $n = 4$; DMSO (0.1%) = 1): 10 µM: $0.93 \pm 0.04$; 30 µM: $1.01 \pm 0.08$; 60 µM: $0.81 \pm 0.06$; 100 µM: $1.37 \pm 0.02$; TRITON X-100: $4.22 \pm 0.12$.

Metabolism Assay: The solubilised test compound 1 (5 µL, final concentration 10 µM in DMSO) was preincubated at 37 °C in 432 µL of phosphate buffer (0.1 M, pH 7.4) together with a 50 µL NADPH regenerating system (30 mM glucose-6-phosphate, 4 U/mL glucose-6-phosphate dehydrogenase, 10 mM NADP, 30 mM $MgCl_2$). After 5 min, the reaction was started by the addition of 13 µL of microsome mix from the liver of Sprague−Dawley rats (Invitrogen; 20 mg protein/ mL in 0.1 M phosphate buffer) in a shaking water bath at 37 °C. The reaction was stopped by addition of 250 µL of ice-cold methanol at 0, 15, 30, and 60 min. The samples were diluted with 250 µL of DMSO and centrifuged at 10000 g for 5 min at 4 °C. The supernatants were analysed, and the test compound was quantified by HPLC: mobile phase, MeOH 83%/$H_2O$ 17%/formic acid 0.1%; flow-rate, 1 mL/ min; stationary phase, MultoHigh Phenyl phase, 5 µm, $250 \times 4$ (CS-Chromatography, Langerwehe, Germany); precolumn, phenyl, 5 µm, $20 \times 4$ (CS-Chromatography); detection wavelength, 330 and 254 nm; injection volume, 50 µL. Control samples were performed to check the stability of the compound in the reaction mixture: first control was without NADPH, which is needed for the enzymatic activity of the microsomes, second control was with inactivated microsomes (incubated for 20 min at 90 °C), third control was without test compound (to determine the baseline). The amounts of the test compound were quantified by external calibration. The metabolism experiments showed the following results (expressed as mean ± SEM% remaining compound; $n = 4$): 0 min: $96 \pm 1\%$; 15 min: $84 \pm 3\%$; 30 min: $77 \pm 2\%$; 60 min: $63 \pm 2\%$[40].

**In vivo pharmacology**. Animals and compound application: 15 male C57BL/6JRj mice (23-26 g body weight, purchased from Janvier Labs, Le Genest-Saint-Isle, France) were used in the present study. The animals were housed in a temperature-controlled room (20–24 °C) and maintained in a 12 h light/12 h dark cycle. Food and water were available ad libitum. The in-life phase was performed by the contract research organization Pharmacelsus (Saarbrücken, Germany). All experimental procedures were approved by and conducted in accordance with the regulations of the local Animal Welfare authorities (Landesamt für Gesundheit und Verbraucherschutz, Abteilung Lebensmittel- und Veterinärwesen, Saarbrücken). Nine animals received a single oral dose of 10 mg/kg body weight of FXR partial agonist 1 in water containing 1% HPMC/Tween 80 (99:1), three animals received 10 mg/kg body weight of FXR agonist CDCA in the vehicle and three animals received the vehicle (water containing 1% HPMC/Tween 80 (99:1)). All animals behaved normal throughout the study and showed no adverse effects.

Sample collection and pharmacokinetic analysis: At three time points (15 min, 60 min, 240 min after application of 1, blood from three restrained and conscious mice was obtained from the lateral tail vein. At another three time points (30, 120, 480 min after application of 1, mice were anesthetised under isoflurane and blood (~500 µl) was obtained by retro-orbital puncture leading to three plasma samples per time point. For liver collection, mice receiving 1 sacrificed at the last time point and mice receiving CDCA or vehicle were sacrificed 480 min after application by cervical dislocation and the complete liver was immediately snap-frozen and stored at −80 °C until further evaluation. 1 was quantified from blood samples by HPLC-MS with a flow rate of 600 µl/min on a Kinetex Phenyl-Hexyl, 2.6 µm, $50 \times 2.1$ mm (Phenomenex, Aschaffenburg, Germany) analytical column with a pre-column (Kinetex Phenyl-Hexyl, SecurityGuard Ultra, 2.1 mm). Gradient elution with water and 0.1% formic acid as aqueous phase (A) and acetonitrile with 0.1% formic acid as organic phase (B) was used: % B (t (min)), 0(0–0.1)–97(0.4–1.7)–0(1.8–3.0). Full scan mass spectra were acquired in the positive ion mode using syringe pump infusion to identify the protonated quasi-molecular ions $[M + H]^+$. Pharmacokinetic analysis was performed applying a non-compartment model

using the Kinetica 5.0 software (Thermo Fisher Scientific, Waltham, MA, U.S.A.). Key pharmacokinetic parameters are shown in Supplementary Table 3.

Quantification of FXR target gene mRNA from mouse livers: Hepatocyte isolation of mouse liver tissue for RT-qPCR: To homogenise the liver samples, one third of each liver was placed on one Falcon$^{TM}$ Cell Strainer with 40 μm pore size (BD Biosciences) in a 50 mL Falcon tube. Every tissue was rinsed with PBS buffer containing 10% FCS, penicillin (100 U/mL) and streptomycin (100 μg/mL), and pressed through the cell strainer until 5 mL cell suspension had been collected. The samples were centrifuged at 1200 rpm for 10 min at 4 °C. The supernatant was discarded, and the pellets were washed with 5 mL cold PBS buffer and again centrifuged at 1200 rpm for 10 min at 4 °C. After discarding the supernatant, the cell pellets were re-suspended in 1 mL PBS and total RNA was extracted using the E. Z. A. Total RNA Kit I (Omega Bio-tek) following the Animal Tissue Protocol. The extracted RNA was used for qRT-PCR and equally treated as described for mRNA quantification from HepG2 cells (see above). PCR primers for the murine genes are shown in Supplementary Table 2. Results were normalised to GAPDH. The relative mRNA expression levels were analysed by the $2^{-\Delta Ct}$ method. Results (mean ± SEM; n = 3): BSEP: vehicle: 0.030 ± 0.011; CDCA (10 mg/kg): 0.248 ± 0.035; **1** (10 mg/kg): 0.129 ± 0.018. CYP7A1: vehicle: 0.0453 ± 0.0200; CDCA (10 mg/kg): 0.0067 ± 0.0005; **1** (10 mg/kg): 0.0176 ± 0.0014.

**Protein expression and purification.** Construct: FXR ligand binding domain (FXR-LBD; isoform 4; amino acids 244–472) and the C-terminally (amino acids 244–442) truncated constructs were cloned in pET (ampicillin resistance) with an N-terminal hexahistidine tag and a Tev cleavage site between FXR-LBD and His-tag. DNA sequence (insert):

GAACTGACCCCAGATCAACAGACTCTTCTACATTTTATTATGGATT CATATAACAAACAGAGGATGCCTCAGGAAATAACAAATAAAATTT TAAAAGAAGAATTCAGTGCAGAAGAAAATTTTCTCATTTTGACG GAAATGGCAACCAATCATGTACAGGTTCTTGTAGAATTCACAAAAAAGC TACCAGGATTTCAGACTTTGGACCATGAAGACCAGATTGCTTTGCT GAAAGGGTCTGCGGTTGAAGCTATGTTCCTTCGTTCAGCTGAGATTTT CAATAAGAAACTTCCGTCTGGGCATTCTGACCTATTGGAAGAAA GAATTCAAAATAGTGGTATCTCTGATGAATATATAACACCTATGT TAGTTTTTATAAAAGTATTGGGGAACTGAAAATGACTCAAGAGGAG TATGCTCTGCTTACAGCAATTGTTATCCTGTCTCCAGATAGACAATACA TAAAGGATAGAGAGGCAGTAGAGAAGCTTCAGGAGCCACTTCTTGAT GTGCTACAAAAGTTGTGTAAGATTCACCAGCCTGAAAATCCTCAA CACTTTGCCTGTCTCCTGGGTCGCCTGACTGAATTACGGACATTCAAT CATCACCACGCTGAGATGCTGATGTCATGGAGAGTAAACGACCA CAAGTTTACCCCACTTCTCTGTGAAATCTGGGACGTGCAGTGA

Protein sequence:
ELTPDQQTLLHFIMDSYNKQRMPQEITNKILKEEFSAEENFLILTEMATNH VQVLVEFTKKLPGFQTLDHEDQIALLKGSAVEAMFLRSAEIFNKKLPSGHSDL-LEERIRNSGISDEYITPMFSFYKSIGELKMTQEEYALLTAIVILSPDRQYIKDREA-VEKLQEPLLDVLQKLCKIHQPENPQHFACLLGRLTELRTFNHHHAEMLMSW RVNDHKFTPLLCEIWDVQ

FXR-LBD was expressed in *E. coli* BL21(DE3) gold cells (Invitrogen) in TB medium containing ampicillin in shaking flasks. Cells were initially grown at 37 °C and then continuously cooled to reach 18 °C when OD600 reaches 1. Expression was then induced with IPTG (final concentration 0.1 mM), cells were grown overnight and then harvested. Pellets were stored at −80 °C or directly used for purification.

Protein purification: To the pellet of 1 L culture (~16 g), 60 mL lysis buffer (20 mM Tris pH 7.5, 500 mM NaCl, 1 mM TCEP, 10% glycerol and 2 EDTA-free inhibitor tablets) were added. Cells were disrupted by sonication on ice and then centrifuged at 12,000 rpm for 30 min. The cleared supernatant was treated with imidazole (final concentration 20 mM) and incubated with 10 mL NiNTA. After 1 h incubation, protein was washed with additional lysis buffer containing 20 mM imidazole before product was eluted with elution buffer (250 mM imidazole pH 7.5, 200 mM NaCl, 5 mM TCEP, 10% glycerol). Elute was treated with 500 U Tev and dialysed against 50 mM Tris pH 8.3, 10% glycerol, 100 mM NaCl, 10 mM DTT overnight. The sample was then concentrated to 10 mL and filtrated over Superdex 200 26/60 (GE Healthcare, Chicago, IL, U.S.A.) in 50 mM Tris pH 8.3, 500 mM NaCl, 10 mM DTT, 10 % glycerol. Pure fractions with FXR-LBD monomer were collected and dialysed against 10 mM Tris pH 8.3, 100 mM NaCl, 5 mM DTT at 4 °C overnight to yield 20–50 mg FXR-LBD of >99% (SDS-PAGE) purity per liter cell culture. MS(ESI): calculated: 27181.2 Da; found: 27184.0 Da. Functionality was tested by ITC with GW4064. Mutants were prepared by designing an appropriate primer with the single amino acid mutation and following the standard QuickChange® site-directed mutagenesis protocol. Expression and purification procedure was performed as described for the wild type.

**Synthesis of co-activator/co-repressor peptide.** Co-activator (NCoA; KENALLRYLLDKD) and co-repressor (NCoR; DPASNLGLE-DIIRKALMGSFDDK) were synthesised with free amino and carboxy terminus using solid phase peptide synthesis on an ABI 433 A peptide synthesizer (Thermo Fisher). The resulting product was dissolved in water and further purified by reversed phase HPLC with C18 column material, followed by determination of purity and identity of the product by NMR and ESI-MS.

**NMR spectroscopy.** NMR experiments were performed at 298 K, on Bruker 800 MHz spectrometers equipped with cryogenic triple-resonance probes. Typically, NMR samples contained 0.2 ml of up to 50/100 μM protein in 50 mM Tris pH 8.3, 500 mM NaCl, 2 mM TCEP, 1 mM TMSP, 5% D$_2$O. The spectra were processed and analysed using Topspin 3.5 (Bruker Biospin). For acquiring the $^1$H, $^{13}$C/$^{15}$N-HSQC spectra of the FXR –peptide (NCoA or NCoR) complex, a ratio of 1:1 (50 μM protein; 50 μM peptide) was used for the measurements. The direct interaction of ligands (agonist/antagonist/partial agonist) and their influence on the binding of the co-activator/co-repressor binding was monitored by measuring $^1$H/$^{13}$C-HSQC spectra of the $^{13}$C-$^{15}$N-Arg/Leu selectively labeled peptides (ratio: Protein:peptide: ligand was 50:50:250 μM) in the apo and in complex with protein. Samples were measured in a 3 mm NMR-tube with a sample volume of 0.2 mL. FXR was at a concentration of 50 μM and the peptide and ligands were added to a final concentration of 50 μM (from a stock solution of 1 mM in water; the stock solutions were preheated to 60–70 °C for 10 min to dissolve any aggregates) and 250 μM (from a stock solution of 100 mM in d$_6$-DMSO), respectively. (Note: most ligands were not completely soluble. The insoluble precipitate of the excess ligand was spun down and the clear sample solution was then used for NMR measurements).

**Protein crystallisation and structure determination.** FXR244-472 (75 μM) was incubated on ice for 1 h with 150 μM Iigands (**1** or **2**). The complex was concentrated up to 375 μM and incubated for an additional hour with 750 μM in-house synthesised Nuclear receptor co-activator 2 peptide (NCoA-2740-752). The fresh samples were crystallised at 4 °C as hanging drop in 50 mM Ca(CH$_3$COO)$_2$ pH 7.5, 10 mM DTT, 25-30% PEG 3350 (CDCA) or 300 mM MgCl$_2$, 100 mM Bis-Tris pH 7, 15-20% PEG 3350 (DM175). FXR/CDCA crystals appeared within short period, were soaked in 20% ethylene glycol and flash-frozen in liquid nitrogen. Initial FXR/DM175 crystals diffracted weak and were improved by micro seeding. Diffraction data have been collected on I03 MX at the Diamond Light Source (Didcot, Oxfordshire, UK) and BL14.2 operated by the Joint Berlin MX-Laboratory at the BESSY II electron storage ring (Berlin-Adlershof, Germany)[41]. Data was processed using autoPROC[42] or XDSAPP[43]. Structures were determined by molecular replacement utilising pdb 3FLI as model structure and refined using PHENIX[44].

**Analysis of nuclear receptor LBD structures from the PDB.** InterPro (protein sequence analysis and classification) was used to search for the structures pertaining to the LBD using the code IPR000536. The resulting PDB codes were read to extract the amino acid sequence from each chain and multiple sequence alignments were performed using Kalign. The residues Lys (α3) and Glu (α12) are widely conserved and typically form a salt bridge in the ligand-bound state of the LBD. D1 indicates the distance between Cα of both residues and was extracted for each of the chains within the PDB. In total there were 578 structures with peptide (co-activator/co-repressor) bound and 740 structures without peptide. The D1 distances extracted from these structures were plotted against the PDB.

**Reporting summary.** Further information on research design is available in the Nature Research Reporting Summary linked to this article.

## Data availability
Macromolecular data has been deposited in the PDB with the accession codes 4QE8 [https://doi.org/10.2210/pdb4QE8/pdb], 6HL0 [https://doi.org/10.2210/pdb6HL0/pdb], and 6HL1 [https://doi.org/10.2210/pdb6HL1/pdb]. The source data underlying Fig. 1, Table 1, Supplementary Fig. 4 and Supplementary Fig. 6 are provided as a Source Data file. Other data are available from the corresponding authors on reasonable request.

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

## Acknowledgements

This work has been supported by iNEXT, grant number 653706, funded by the Horizon 2020 programme of the European Commission. This research was financially supported by the German Cancer Consortium (DKTK) and DFG (SFB807).

## Author contributions

D.M., S.S., D.K., K.S., V.L., S.L.G., F.H. and C.L. performed the experiments. E.N., A.A., L.W., N.D. and K.B. contributed protein, expression construct and protocols. D.M., S.S., M.S.-Z. and H.S. and in part D.K and K.S. designed the study. D.M., S.S., M.S.-Z. and H.S. supervised the study. D.M., S.S., M.S.-Z. and H.S. wrote the manuscript.

## Additional information

**Competing interests:** The authors declare no competing interests.

