## [Peer Review File · Nature Communications]

Reviewers' comments:

Reviewer #1 (Remarks to the Author):

This is a revised manuscript describing structural mechanisms of FXR partial agonism. Overall, this is an interesting and timely study. The authors have addressed some of the concerns from the previous review, but a few concerns remain:

The authors added new luciferase assay data (Figure 1c) where the FXR LBD is fused to the Gal4 DNA-binding domain. Compound 1 activates this Gal4-FXR LBD construct by ~1-fold over what would be expected for vehicle control (relative activation values of 0.1 at low ligand concentration vs. 0.2 at high ligand concentration). The authors provide these data to compare to Gal4-fusions of several other NR LBDs (Figure 1d) to indicate Compound 1 has selectivity over other lipid-binding nuclear receptors. However, Compound 1 activates several other non-FXR Gal4-NRs to a larger degree than Gal4-FXR; e.g. Gal4-PPAR γ and Gal4-RAR α = 5-fold increase in activation by Compound 1, and many others seem to have fold activities greater than or equal to the 1-fold increase for Gal4-FXR. This indicates Compound 1 may not be a specific FXR partial agonist. The authors should graphically compare a Gal4-FXR "fold activation" comparison to the non-FXR NRs (e.g., as shown in Figure 1d to directly compare to non-FXR NRs) and comment on the lack of apparent FXR specificity.

Also related to the new Gal4-FXR data (Figure 1c), Compound 1 elicits a $>1 \mu\text{M}$ EC₅₀ alone but in the GW4064 competition assay shows a ~100 nM EC₅₀—more potent than when Compound 1 is treated alone, which is opposite of what I would have expected when competing with a higher affinity agonist. Why is there such a large discrepancy?

The authors included new mutagenesis data to support a functional role for W454. However, the W454 side chain density in DM175 bound structure (PDB 4QE8) is still not convincing enough to conclude the W454 side chain is 100% fitted correctly. Please see the accompanying image, where the density around W454 in chain B is contoured at level 1.0 for 2Fo-Fc and 2.7 for Fo-Fc. W454 is the last residue modeled, but the remaining unmodeled density in this region makes the placement of the W454 side chain unclear as it could potentially correspond to additional backbone density for other residues not modeled into the structure.

In Table 1, the W454 mutants did not affect GW4064 activity, but did have an effect on CDCA and Compound 1. Can the authors speculate into the structural mechanism for the lack of activity on GW4064 based on the W454 and GW4064 conformations in published crystal structures (e.g. PDB 3DCT)? Also, the dose response curves used to fit these data should be shown in the Supplemental Data document for complete transparency.

In the crystal structure of Compound 1 (chain B), since the R455 was not modeled the authors should correct this sentence on line 156 to start at R455: "Additionally, the electron density of the loop region connecting α 11 and α 12 (V456-H459, AF-2 loop) is ambiguous or invisible, due to induced flexibility by binding of 1."

The NMR overlays in Figure 4 are difficult to compare for two reasons: there are 3 different spectra overlaid in each panel and the choice of colors (blue, red, teal) do not make the changes obvious. I recommend only showing overlays of 2 spectra per panel. Although this will increase the number of panels shown, it will make it easier for the readers to see what the authors describe in the text.

The author's response about the "real-time" nature of the NMR-based peptide binding assay to assess how ligands affect the selectivity for corepressor and coactivator binding only provides a qualitative assessment. There are more quantitative ways to show how the ligands affect corepressor and coactivator binding affinities directly, such as ITC or fluorescence polarization assays, which would provide more compelling support (affinities) for coregulator peptide

selectivity.

This is the first report of Compound 1. NMR chemical shift values (^1H and ^{13}C) and mass determined from HRMS data are provided in the Methods section. However, the actual NMR spectra should also be shown in the Supplemental Data document.

H12 and α 12 are both used to describe helix 12, for example. One type of nomenclature should be used consistently throughout.

There are two different Figure 4s shown. Please be sure to update the Figure numbering.

Dear Reviewers

Thank you for the very careful review of our manuscript and for further constructive suggestions to improve its quality. We have addressed all your suggestions and concerns. Point-by-point answers are listed below.

We hope this revised version meets your expectations and we are looking forward to your response.

Reviewer #1 (Remarks to the Author):

This is a revised manuscript describing structural mechanisms of FXR partial agonism. Overall, this is an interesting and timely study. The authors have addressed some of the concerns from the previous review, but a few concerns remain:

Thank you very much for your positive response and your further suggestions to improve the manuscript! We have followed up on all suggestions and are convinced that our main conclusions are even strengthened after having conducted the additional experiments.

The authors added new luciferase assay data (Figure 1c) where the FXR LBD is fused to the Gal4 DNA-binding domain. Compound 1 activates this Gal4-FXR LBD construct by ~1-fold over what would be expected for vehicle control (relative activation values of 0.1 at low ligand concentration vs. 0.2 at high ligand concentration). The authors provide these data to compare to Gal4-fusions of several other NR LBDs (Figure 1d) to indicate Compound 1 has selectivity over other lipid-binding nuclear receptors. However, Compound 1 activates several other non-FXR Gal4-NRs to a larger degree than Gal4-FXR; e.g. Gal4-PPAR γ and Gal4-RAR α = 5-fold increase in activation by Compound 1, and many others seem to have fold activities greater than or equal to the 1-fold increase for Gal4-FXR. This indicates Compound 1 may not be a specific FXR partial agonist. The authors should graphically compare a Gal4-FXR “fold activation” comparison to the non-FXR NRs (e.g., as shown in Figure 1d to directly compare to non-FXR NRs) and comment on the lack of apparent FXR specificity.

We thank the reviewer for this careful analysis of our data and have revised the manuscript accordingly: We agree with the reviewer that displaying relative activation for the dose-response curves (Figure 1b and c) but fold activation for the selectivity screening (Figure 1d) is not well comparable and we have adapted Figure 1d accordingly for better understanding. We have chosen to stick to relative activation for all reporter gene assay data for two reasons. First, as the focus of this manuscript is partial agonism, relative activation compared to a full agonist gives a better picture of the partial agonistic profile than fold activation. Second, nuclear receptors (e.g. FXR vs RAR) differ in their intrinsic/baseline activity leading to broad variance of the fold activation of full agonists on distinct NRs. E.g. the full RAR agonist tretinoin has a much stronger fold activation efficacy than the FXR agonist GW4064 which in part is due to the higher intrinsic/baseline activity of FXR. Displaying relative activation normalizes all activity values to a range between 0 and 1 (0% and 100%) which provides a better comparability. As an additional advantage, in fold activation, the negative control data (DMSO 0.1%) is defined as 1 and has no standard deviation by definition while it has variance in relative activation where the positive control data (reference agonist) is defined as 1. Therefore, relative activation also allows a more reasonable statistical analysis of the data.

Also related to the new Gal4-FXR data (Figure 1c), Compound 1 elicits a $>1 \mu\text{M}$ EC₅₀ alone but in the GW4064 competition assay shows a $\sim 100 \text{ nM}$ EC₅₀—more potent than when Compound 1 is treated alone, which is opposite of what I would have expected when competing with a higher affinity agonist. Why is there such a large discrepancy?

We appreciate this comment by the reviewer which relies on the careful analysis of our data. We address this as follows: the Gal4 hybrid reporter gene assays constitute a model system that reflects nuclear receptor activity only partially. Nevertheless, it is used not only by us, but also in the literature as activity reporter (e.g. Solt, L. et al. Nature 2011, 472:491-4). The chimeric receptors that govern reporter activity in these assays may be compromised in their ability to recruit different coactivators and differentially form monomers or dimers (e.g. Hong, M. et al. Structure 2008, 16:1019-26). Moreover, the response sequence for DNA binding of the hybrid receptor can accommodate several receptor monomers/dimers at the same time which can further affect the results. Thus, we agree with the reviewer that the higher antagonistic potency of the partial agonist in the hybrid assay system is unexpected but may arise from a variety of effects caused by the hybrid assay setup. We are fully convinced by our full length assay data since this test system better reflects the physiological hetero-dimer situation. In addition, the hybrid test system reproduced the partial agonist activity of DM175 well compared to the full length assay and only revealed a discrepancy concerning the competitive activity.

The authors included new mutagenesis data to support a functional role for W454. However, the W454 side chain density in DM175 bound structure (PDB 4QE8) is still not convincing enough to conclude the W454 side chain is 100% fitted correctly. Please see the accompanying image, where the density around W454 in chain B is contoured at level 1.0 for 2Fo-Fc and 2.7 for Fo-Fc. W454 is the last residue modeled, but the remaining unmodeled density in this region makes the placement of the W454 side chain unclear as it could potentially correspond to additional backbone density for other residues not modeled into the structure.

We thank the reviewer for this further careful analysis of our structural data. Indeed, sometimes it is challenging to perfectly assign and place amino acid side chains like W454 or R455 in flexible loops at 2.6 Angstroms resolution with limited data redundancy resulting from low symmetry P1 space group.

Further in-depth analysis of ligand binding indicates DM175 occupancies of about 85% in chain A and 75% in chain B, respectively. This might allow for a minor fraction of unbound FXR molecules possibly having altered loop conformations, leading to more flexibility. Still, our data indicate that the major fraction of W454 is pointing outwards of the ligand binding pocket.

Furthermore, the whole loop in chain B containing W454 is significantly more distorted than in chain A. Although NCS was taken in account for data refinement with Phenix, the electron density did not improve. It would be possible to place W454 in chain A adopting an alternative conformation, but this would not affect our main structural conclusions. The presented fitting of W454 in chain B is the most preferred structural model after refinement with Phenix AND CCP4. This conformation shows a reasonable density fit and no statistical outliers or clashes.

It is indeed possible to place R455 (chain B) into the residual unmodeled density, but subsequent data refinement resulted either in clashing with A200 (B), S355 (B) or G356

(B), violated Ramachandran and rotamer regulations, or led to disruption of the covalent constitution of the protein.

In Table 1, the W454 mutants did not affect GW4064 activity, but did have an effect on CDCA and Compound 1. Can the authors speculate into the structural mechanism for the lack of activity on GW4064 based on the W454 and GW4064 conformations in published crystal structures (e.g. PDB 3DCT)? Also, the dose response curves used to fit these data should be shown in the Supplemental Data document for complete transparency.

We thank the reviewer for this suggestion and have updated the manuscript with further discussion of this aspect. The FXR-LBD co-crystals in complex with GW4064 and CDCA indeed point to an explanation of the different effects of W454 mutations on the activity of both ligands. We have added a paragraph addressing this matter in the discussion section accompanied by figure S8 in the SI. Moreover, we have included the dose-response curves in Figure S4.

In the crystal structure of Compound 1 (chain B), since the R455 was not modeled the authors should correct this sentence on line 156 to start at R455: “Additionally, the electron density of the loop region connecting $\alpha 11$ and $\alpha 12$ (V456-H459, AF-2 loop) is ambiguous or invisible, due to induced flexibility by binding of 1.”

We thank the reviewer for this and have added the suggested wording.

The NMR overlays in Figure 4 are difficult to compare for two reasons: there are 3 different spectra overlaid in each panel and the choice of colors (blue, red, teal) do not make the changes obvious. I recommend only showing overlays of 2 spectra per panel. Although this will increase the number of panels shown, it will make it easier for the readers to see what the authors describe in the text.

We thank the reviewer for this suggestion. The figure has been modified to show two overlays for each FXR ligand as suggested.

The author’s response about the “real-time” nature of the NMR-based peptide binding assay to assess how ligands affect the selectivity for corepressor and coactivator binding only provides a qualitative assessment. There are more quantitative ways to show how the ligands affect corepressor and coactivator binding affinities directly, such as ITC or fluorescence polarization assays, which would provide more compelling support (affinities) for coregulator peptide selectivity.

The reviewer comments on our response in the previous cover letter. We completely accept his point, the expression “real-time” nature of the NMR-based peptide binding assay is misleading.

We agree with the reviewer on the strength of ITC in characterizing these interactions. In fact, we tried to conduct less demanding MST experiments but given the stability of protein and the solubility of the hydrophobic ligands made these experiments unsuccessful. With regard to fluorescence anisotropy measurements and changes upon binding, in particular as repressor and co-activator peptides have been added, we are not convinced that these experiments would actually generate additional data.

This is the first report of Compound 1. NMR chemical shift values (^1H and ^{13}C) and mass determined from HRMS data are provided in the Methods section. However, the actual NMR spectra should also be shown in the Supplemental Data document.

The requested NMR spectra have been added to the SI.

H12 and α 12 are both used to describe helix 12, for example. One type of nomenclature should be used consistently throughout.

We thank the reviewer for carefully reading our manuscript. Now only “ α ” is used to describe helices throughout the manuscript.

There are two different Figure 4s shown. Please be sure to update the Figure numbering.

Figure numbering has been checked carefully and corrected as appropriate.